# Revisiting Perceptron:
# Efficient and Label-Optimal Learning of Halfspaces

**Songbai Yan**
UC San Diego
La Jolla, CA
yansongbai@ucsd.edu

**Chicheng Zhang**[*]
Microsoft Research
New York, NY
chicheng.zhang@microsoft.com

## Abstract

It has been a long-standing problem to efficiently learn a halfspace using as few labels as possible in the presence of noise. In this work, we propose an efficient Perceptron-based algorithm for actively learning homogeneous halfspaces under the uniform distribution over the unit sphere. Under the bounded noise condition [49], where each label is flipped with probability at most $\eta < \frac{1}{2}$, our algorithm achieves a near-optimal label complexity of $\tilde{O}\left(\frac{d}{(1-2\eta)^2}\ln\frac{1}{\epsilon}\right)^2$ in time $\tilde{O}\left(\frac{d^2}{\epsilon(1-2\eta)^3}\right)$. Under the adversarial noise condition [6, 45, 42], where at most a $\tilde{\Omega}(\epsilon)$ fraction of labels can be flipped, our algorithm achieves a near-optimal label complexity of $\tilde{O}\left(d\ln\frac{1}{\epsilon}\right)$ in time $\tilde{O}\left(\frac{d^2}{\epsilon}\right)$. Furthermore, we show that our active learning algorithm can be converted to an efficient passive learning algorithm that has near-optimal sample complexities with respect to $\epsilon$ and $d$.

## 1 Introduction

We study the problem of designing efficient noise-tolerant algorithms for actively learning homogeneous halfspaces in the streaming setting. We are given access to a data distribution from which we can draw unlabeled examples, and a noisy labeling oracle $\mathcal{O}$ that we can query for labels. The goal is to find a computationally efficient algorithm to learn a halfspace that best classifies the data while making as few queries to the labeling oracle as possible.

Active learning arises naturally in many machine learning applications where unlabeled examples are abundant and cheap, but labeling requires human effort and is expensive. For those applications, one natural question is whether we can learn an accurate classifier using as few labels as possible. Active learning addresses this question by allowing the learning algorithm to sequentially select examples to query for labels, and avoid requesting labels which are less informative, or can be inferred from previously-observed examples.

There has been a large body of work on the theory of active learning, showing sharp distribution-dependent label complexity bounds [21, 11, 34, 27, 35, 46, 60, 41]. However, most of these general active learning algorithms rely on solving empirical risk minimization problems, which are computationally hard in the presence of noise [5].

On the other hand, existing computationally efficient algorithms for learning halfspaces [17, 29, 42, 45, 6, 23, 7, 8] are not optimal in terms of label requirements. These algorithms have different degrees of noise tolerance (e.g. adversarial noise [6], malicious noise [43], random classification noise [3],

---

[*]Work done while at UC San Diego.
[2]We use $\tilde{O}(f(\cdot)) := O(f(\cdot)\ln f(\cdot))$, and $\tilde{\Omega}(f(\cdot)) := \Omega(f(\cdot)/\ln f(\cdot))$. We say $f(\cdot) = \tilde{\Theta}(g(\cdot))$ if $f(\cdot) = \tilde{O}(g(\cdot))$ and $f(\cdot) = \tilde{\Omega}(g(\cdot))$

bounded noise [49], etc), and run in time polynomial in $\frac{1}{\epsilon}$ and $d$. Some of them naturally exploit the utility of active learning [6, 7, 8], but they do not achieve the sharpest label complexity bounds in contrast to those computationally-inefficient active learning algorithms [10, 9, 60].

Therefore, a natural question is: is there any active learning halfspace algorithm that is computationally efficient, and has a minimum label requirement? This has been posed as an open problem in [50]. In the realizable setting, [26, 10, 9, 56] give efficient algorithms that have optimal label complexity of $\tilde{O}(d \ln \frac{1}{\epsilon})$ under some distributional assumptions. However, the challenge still remains open in the nonrealizable setting. It has been shown that learning halfspaces with agnostic noise even under Gaussian unlabeled distribution is hard [44]. Nonetheless, we give an affirmative answer to this question under two moderate noise settings: bounded noise and adversarial noise.

## 1.1 Our Results

We propose a Perceptron-based algorithm, ACTIVE-PERCEPTRON, for actively learning homogeneous halfspaces under the uniform distribution over the unit sphere. It works under two noise settings: bounded noise and adversarial noise. Our work answers an open question by [26] on whether Perceptron-based active learning algorithms can be modified to tolerate label noise.

In the $\eta$-*bounded noise setting* (also known as the Massart noise model [49]), the label of an example $x \in \mathbb{R}^d$ is generated by $\text{sign}(u \cdot x)$ for some underlying halfspace $u$, and flipped with probability $\eta(x) \leq \eta < \frac{1}{2}$. Our algorithm runs in time $\tilde{O}\left(\frac{d^2}{(1-2\eta)^3\epsilon}\right)$, and requires $\tilde{O}\left(\frac{d}{(1-2\eta)^2} \cdot \ln \frac{1}{\epsilon}\right)$ labels. We show that this label complexity is *nearly optimal* by providing an almost matching information-theoretic lower bound of $\Omega\left(\frac{d}{(1-2\eta)^2} \cdot \ln \frac{1}{\epsilon}\right)$. Our time and label complexities substantially improve over the state of the art result of [8], which runs in time $\tilde{O}(d^{O\left(\frac{1}{(1-2\eta)^4}\right)} \frac{1}{\epsilon})$ and requires $\tilde{O}(d^{O\left(\frac{1}{(1-2\eta)^4}\right)} \ln \frac{1}{\epsilon})$ labels.

Our main theorem on learning under bounded noise is as follows:

**Theorem 2** (Informal). *Suppose the labeling oracle $\mathcal{O}$ satisfies the $\eta$-bounded noise condition with respect to $u$, then for ACTIVE-PERCEPTRON, with probability at least $1-\delta$: (1) The output halfspace $v$ is such that $\mathbb{P}[\text{sign}(v \cdot X) \neq \text{sign}(u \cdot X)] \leq \epsilon$; (2) The number of label queries to oracle $\mathcal{O}$ is at most $\tilde{O}\left(\frac{d}{(1-2\eta)^2} \cdot \ln \frac{1}{\epsilon}\right)$; (3) The number of unlabeled examples drawn is at most $\tilde{O}\left(\frac{d}{(1-2\eta)^3\epsilon}\right)$; (4) The algorithm runs in time $\tilde{O}\left(\frac{d^2}{(1-2\eta)^3\epsilon}\right)$.*

In addition, we show that our algorithm also works in a more challenging setting, the $\nu$-*adversarial noise setting* [6, 42, 45].[3] In this setting, the examples still come iid from a distribution, but the assumption on the labels is just that $\mathbb{P}[\text{sign}(u \cdot X) \neq Y] \leq \nu$ for some halfspace $u$. Under this assumption, the Bayes classifier may not be a halfspace. We show that our algorithm achieves an error of $\epsilon$ while tolerating a noise level of $\nu = \Omega\left(\frac{\epsilon}{\ln \frac{d}{\delta} + \ln \ln \frac{1}{\epsilon}}\right)$. It runs in time $\tilde{O}\left(\frac{d^2}{\epsilon}\right)$, and requires only $\tilde{O}\left(d \cdot \ln \frac{1}{\epsilon}\right)$ labels which is near-optimal. ACTIVE-PERCEPTRON has a label complexity bound that matches the state of the art result of [39][4], while having a lower running time.

Our main theorem on learning under adversarial noise is as follows:

**Theorem 3** (Informal). *Suppose the labeling oracle $\mathcal{O}$ satisfies the $\nu$-adversarial noise condition with respect to $u$, where $\nu < \Theta(\frac{\epsilon}{\ln \frac{d}{\delta} + \ln \ln \frac{1}{\epsilon}})$. Then for ACTIVE-PERCEPTRON, with probability at least $1-\delta$: (1) The output halfspace $v$ is such that $\mathbb{P}[\text{sign}(v \cdot X) \neq \text{sign}(u \cdot X)] \leq \epsilon$; (2) The number of label queries to oracle $\mathcal{O}$ is at most $\tilde{O}\left(d \cdot \ln \frac{1}{\epsilon}\right)$; (3) The number of unlabeled examples drawn is at most $\tilde{O}\left(\frac{d}{\epsilon}\right)$; (4) The algorithm runs in time $\tilde{O}\left(\frac{d^2}{\epsilon}\right)$.*

Table 1: A comparison of algorithms for active learning of halfspaces under the uniform distribution, in the $\eta$-bounded noise model.

| Algorithm | Label Complexity | Time Complexity |
|---|---|---|
| [10, 9, 60] | $\tilde{O}\big(\frac{d}{(1-2\eta)^2}\ln\frac{1}{\epsilon}\big)$ | superpoly$(d,\frac{1}{\epsilon})$ [5] |
| [8] | $\tilde{O}(d^{O(\frac{1}{(1-2\eta)^4})}\cdot\ln\frac{1}{\epsilon})$ | $\tilde{O}(d^{O(\frac{1}{(1-2\eta)^4})}\cdot\frac{1}{\epsilon})$ |
| Our Work | $\tilde{O}\big(\frac{d}{(1-2\eta)^2}\ln\frac{1}{\epsilon}\big)$ | $\tilde{O}\left(\frac{d^2}{(1-2\eta)^3}\frac{1}{\epsilon}\right)$ |

Table 2: A comparison of algorithms for active learning of halfspaces under the uniform distribution, in the $\nu$-adversarial noise model.

| Algorithm | Noise Tolerance | Label Complexity | Time Complexity |
|---|---|---|---|
| [60] | $\nu = \Omega(\epsilon)$ | $\tilde{O}(d\ln\frac{1}{\epsilon})$ | superpoly$(d,\frac{1}{\epsilon})$ |
| [39] | $\nu = \Omega(\epsilon)$ | $\tilde{O}(d\ln\frac{1}{\epsilon})$ | poly$(d,\frac{1}{\epsilon})$ |
| Our Work | $\nu = \Omega(\frac{\epsilon}{\ln d+\ln\ln\frac{1}{\epsilon}})$ | $\tilde{O}(d\ln\frac{1}{\epsilon})$ | $\tilde{O}\left(d^2\cdot\frac{1}{\epsilon}\right)$ |

Throughout the paper, ACTIVE-PERCEPTRON is shown to work if the unlabeled examples are drawn uniformly from the unit sphere. The algorithm and analysis can be easily generalized to any spherical symmetrical distributions, for example, isotropic Gaussian distributions. They can also be generalized to distributions whose densities with respect to uniform distribution are bounded away from 0.

In addition, we show in Section 6 that ACTIVE-PERCEPTRON can be converted to a passive learning algorithm, PASSIVE-PERCEPTRON, that has near optimal sample complexities with respect to $\epsilon$ and $d$ under the two noise settings. We defer the discussion to the end of the paper.

## 2 Related Work

**Active Learning.**   The recent decades have seen much success in both theory and practice of active learning; see the excellent surveys by [54, 37, 25]. On the theory side, many label-efficient active learning algorithms have been proposed and analyzed. An incomplete list includes [21, 11, 34, 27, 35, 46, 60, 41]. Most algorithms relies on solving empirical risk minimization problems, which are computationally hard in the presence of noise [5].

**Computational Hardness of Learning Halfspaces.**   Efficient learning of halfspaces is one of the central problems in machine learning [22]. In the realizable case, it is well known that linear programming will find a consistent hypothesis over data efficiently. In the nonrealizable setting, however, the problem is much more challenging.

A series of papers have shown the hardness of learning halfspaces with agnostic noise [5, 30, 33, 44, 23]. The state of the art result [23] shows that under standard complexity-theoretic assumptions, there exists a data distribution, such that the best linear classifier has error $o(1)$, but no polynomial time algorithms can achieve an error at most $\frac{1}{2}-\frac{1}{d^c}$ for every $c>0$, even with improper learning. [44] shows that under standard assumptions, even if the unlabeled distribution is Gaussian, any agnostic halfspace learning algorithm must run in time $(\frac{1}{\epsilon})^{\Omega(\ln d)}$ to achieve an excess error of $\epsilon$. These results indicate that, to have nontrivial guarantees on learning halfspaces with noise in polynomial time, one has to make additional assumptions on the data distribution over instances and labels.

**Efficient Active Learning of Halfspaces.**   Despite considerable efforts, there are only a few halfspace learning algorithms that are both computationally-efficient and label-efficient even under the uniform distribution. In the realizable setting, [26, 10, 9] propose computationally efficient active learning algorithms which have an optimal label complexity of $\tilde{O}(d\ln\frac{1}{\epsilon})$.

Since it is believed to be hard for learning halfspaces in the general agnostic setting, it is natural to consider algorithms that work under more moderate noise conditions. Under the bounded noise

setting [49], the only known algorithms that are both label-efficient and computationally-efficient are [7, 8]. [7] uses a margin-based framework which queries the labels of examples near the decision boundary. To achieve computational efficiency, it adaptively chooses a sequence of hinge loss minimization problems to optimize as opposed to directly optimizing the 0-1 loss. It works only when the label flipping probability upper bound $\eta$ is small ($\eta \leq 1.8 \times 10^{-6}$). [8] improves over [7] by adapting a polynomial regression procedure into the margin-based framework. It works for any $\eta < 1/2$, but its label complexity is $O(d^{O(\frac{1}{(1-2\eta)^4})} \ln \frac{1}{\epsilon})$, which is far worse than the information-theoretic lower bound $\Omega(\frac{d}{(1-2\eta)^2} \ln \frac{1}{\epsilon})$. Recently [20] gives an efficient algorithm with a near-optimal label complexity under the membership query model where the learner can query on synthesized points. In contrast, in our stream-based model, the learner can only query on points drawn from the data distribution. We note that learning in the stream-based model is harder than in the membership query model, and it is unclear how to transform the DC algorithm in [20] into a computationally efficient stream-based active learning algorithm.

Under the more challenging $\nu$-adversarial noise setting, [6] proposes a margin-based algorithm that reduces the problem to a sequence of hinge loss minimization problems. Their algorithm achieves an error of $\epsilon$ in polynomial time when $\nu = \Omega(\epsilon)$, but requires $\tilde{O}(d^2 \ln \frac{1}{\epsilon})$ labels. Later, [39] performs a refined analysis to achieve a near-optimal label complexity of $\tilde{O}(d \ln \frac{1}{\epsilon})$, but the time complexity of the algorithm is still an unspecified high order polynomial.

Tables 1 and 2 present comparisons between our results and results most closely related to ours in the literature. Due to space limitations, discussions of additional related work are deferred to Appendix A.

# 3  Definitions and Settings

We consider learning homogeneous halfspaces under uniform distribution. The instance space $\mathcal{X}$ is the unit sphere in $\mathbb{R}^d$, which we denote by $\mathbb{S}^{d-1} := \{x \in \mathbb{R}^d : \|x\| = 1\}$. We assume $d \geq 3$ throughout this paper. The label space $\mathcal{Y} = \{+1, -1\}$. We assume all data points $(x, y)$ are drawn i.i.d. from an underlying distribution $D$ over $\mathcal{X} \times \mathcal{Y}$. We denote by $D_{\mathcal{X}}$ the marginal of $D$ over $\mathcal{X}$ (which is uniform over $\mathbb{S}^{d-1}$), and $D_{Y|X}$ the conditional distribution of $Y$ given $X$. Our algorithm is allowed to draw unlabeled examples $x \in \mathcal{X}$ from $D_{\mathcal{X}}$, and to make queries to a labeling oracle $\mathcal{O}$ for labels. Upon query $x$, $\mathcal{O}$ returns a label $y$ drawn from $D_{Y|X=x}$. The hypothesis class of interest is the set of homogeneous halfspaces $\mathcal{H} := \{h_w(x) = \text{sign}(w \cdot x) \mid w \in \mathbb{S}^{d-1}\}$. For any hypothesis $h \in \mathcal{H}$, we define its error rate $\text{err}(h) := \mathbb{P}_D[h(X) \neq Y]$. We will drop the subscript $D$ in $\mathbb{P}_D$ when it is clear from the context. Given a dataset $S = \{(X_1, Y_1), \ldots, (X_m, Y_m)\}$, we define the empirical error rate of $h$ over $S$ as $\text{err}_S(h) := \frac{1}{m} \sum_{i=1}^{m} \mathbb{1}\{h(x_i) \neq y_i\}$.

**Definition 1** (Bounded Noise [49]). *We say that the labeling oracle $\mathcal{O}$ satisfies the $\eta$-bounded noise condition for some $\eta \in [0, 1/2)$ with respect to $u$, if for any $x$, $\mathbb{P}[Y \neq \text{sign}(u \cdot x) \mid X = x] \leq \eta$.*

It can be seen that under $\eta$-bounded noise condition, $h_u$ is the Bayes classifier.

**Definition 2** (Adversarial Noise [6]). *We say that the labeling oracle $\mathcal{O}$ satisfies the $\nu$-adversarial noise condition for some $\nu \in [0, 1]$ with respect to $u$, if $\mathbb{P}[Y \neq \text{sign}(u \cdot X)] \leq \nu$.*

For two unit vectors $v_1, v_2$, denote by $\theta(v_1, v_2) = \arccos(v_1 \cdot v_2)$ the angle between them. The following lemma gives relationships between errors and angles (see also Lemma 1 in [8]).

**Lemma 1.** *For any $v_1, v_2 \in \mathbb{S}^{d-1}$, $|\text{err}(h_{v_1}) - \text{err}(h_{v_2})| \leq \mathbb{P}[h_{v_1}(X) \neq h_{v_2}(X)] = \frac{\theta(v_1, v_2)}{\pi}$.*

*Additionally, if the labeling oracle satisfies the $\eta$-bounded noise condition with respect to $u$, then for any vector $v$, $|\text{err}(h_v) - \text{err}(h_u)| \geq (1 - 2\eta)\mathbb{P}[h_v(X) \neq h_u(X)] = \frac{1-2\eta}{\pi}\theta(v, u)$.*

Given access to unlabeled examples drawn from $D_{\mathcal{X}}$ and a labeling oracle $\mathcal{O}$, our goal is to find a polynomial time algorithm $\mathcal{A}$ such that with probability at least $1 - \delta$, $\mathcal{A}$ outputs a halfspace $h_v \in \mathcal{H}$ with $\mathbb{P}[\text{sign}(v \cdot X) \neq \text{sign}(u \cdot X)] \leq \epsilon$ for some target accuracy $\epsilon$ and confidence $\delta$. (By Lemma 1, this guarantees that the excess error of $h_v$ is at most $\epsilon$, namely, $\text{err}(h_v) - \text{err}(h_u) \leq \epsilon$.) The desired algorithm should make as few queries to the labeling oracle $\mathcal{O}$ as possible.

We say an algorithm $\mathcal{A}$ achieves a *label complexity* of $\Lambda(\epsilon, \delta)$, if for any target halfspace $h_u \in \mathcal{H}$, with probability at least $1 - \delta$, $\mathcal{A}$ outputs a halfspace $h_v \in \mathcal{H}$ such that $\mathrm{err}(h_v) \leq \mathrm{err}(h_u) + \epsilon$, and requests at most $\Lambda(\epsilon, \delta)$ labels from oracle $\mathcal{O}$.

## 4   Main Algorithm

Our main algorithm, ACTIVE-PERCEPTRON (Algorithm 1), works in epochs. It works under the bounded and the adversarial noise models, if its sample schedule $\{m_k\}$ and band width $\{b_k\}$ are set appropriately with respect to each noise model. At the beginning of each epoch $k$, it assumes an upper bound of $\frac{\pi}{2^k}$ on $\theta(v_{k-1}, u)$, the angle between current iterate $v_{k-1}$ and the underlying halfspace $u$. As we will see, this can be shown to hold with high probability inductively. Then, it calls procedure MODIFIED-PERCEPTRON (Algorithm 2) to find an new iterate $v_k$, which can be shown to have an angle with $u$ at most $\frac{\pi}{2^{k+1}}$ with high probability. The algorithm ends when a total of $k_0 = \lceil \log_2 \frac{1}{\epsilon} \rceil$ epochs have passed.

For simplicity, we assume for the rest of the paper that the angle between the initial halfspace $v_0$ and the underlying halfspace $u$ is acute, that is, $\theta(v_0, u) \leq \frac{\pi}{2}$; Appendix F shows that this assumption can be removed with a constant overhead in terms of label and time complexities.

---

**Algorithm 1** ACTIVE-PERCEPTRON

---

**Input:** Labeling oracle $\mathcal{O}$, initial halfspace $v_0$, target error $\epsilon$, confidence $\delta$, sample schedule $\{m_k\}$, band width $\{b_k\}$.
**Output:** learned halfspace $v$.
  1: Let $k_0 = \lceil \log_2 \frac{1}{\epsilon} \rceil$.
  2: **for** $k = 1, 2, \ldots, k_0$ **do**
  3:     $v_k \leftarrow$ MODIFIED-PERCEPTRON$(\mathcal{O}, v_{k-1}, \frac{\pi}{2^k}, \frac{\delta}{k(k+1)}, m_k, b_k)$.
  4: **end for**
  5: **return** $v_{k_0}$.

---

Procedure MODIFIED-PERCEPTRON (Algorithm 2) is the core component of ACTIVE-PERCEPTRON. It sequentially performs a modified Perceptron update rule on the selected new examples $(x_t, y_t)$ [51, 17, 26]:

$$w_{t+1} \leftarrow w_t - 2\mathbb{1}\{y_t w_t \cdot x_t < 0\}(w_t \cdot x_t) \cdot x_t \tag{1}$$

Define $\theta_t := \theta(w_t, u)$. Update rule (1) implies the following relationship between $\theta_{t+1}$ and $\theta_t$ (See Lemma 8 in Appendix E for its proof):

$$\cos\theta_{t+1} - \cos\theta_t = -2\mathbb{1}\{y_t w_t \cdot x_t < 0\}(w_t \cdot x_t) \cdot (u \cdot x_t) \tag{2}$$

This motivates us to take $\cos\theta_t$ as our measure of progress; we would like to drive $\cos\theta_t$ up to 1(so that $\theta_t$ goes down to 0) as fast as possible.

To this end, MODIFIED-PERCEPTRON samples new points $x_t$ under time-varying distributions $D_{\mathcal{X}}|_{R_t}$ and query for their labels, where $R_t = \left\{ x \in \mathbb{S}^{d-1} : \frac{b}{2} \leq w_t \cdot x \leq b \right\}$ is a band inside the unit sphere. The rationale behind the choice of $R_t$ is twofold:

1. We set $R_t$ to have a probability mass of $\tilde{\Omega}(\epsilon)$, so that the time complexity of rejection sampling is at most $\tilde{O}(\frac{1}{\epsilon})$ per example. Moreover, in the adversarial noise setting, we set $R_t$ large enough to dominate the noise of magnitude $\nu = \tilde{\Omega}(\epsilon)$.

2. Unlike the active Perceptron algorithm in [26] or other margin-based approaches (for example [55, 10]) where examples with small margin are queried, we query the label of the examples with a range of margin $[\frac{b}{2}, b]$. From a technical perspective, this ensures that $\theta_t$ decreases by a decent amount in expectation (see Lemmas 9 and 10 for details).

Following the insight of [32], we remark that the modified Perceptron update (1) on distribution $D_{\mathcal{X}}|_{R_t}$ can be alternatively viewed as performing stochastic gradient descent on a special non-convex loss function $\ell(w, (x, y)) = \min(1, \max(0, -1 - \frac{2}{b}yw \cdot x))$. It is an interesting open question whether optimizing this new loss function can lead to improved empirical results for learning halfspaces.

**Algorithm 2** MODIFIED-PERCEPTRON

**Input:** Labeling oracle $\mathcal{O}$, initial halfspace $w_0$, angle upper bound $\theta$, confidence $\delta$, number of iterations $m$, band width $b$.

**Output:** Improved halfspace $w_m$.

1: **for** $t = 0, 1, 2, \ldots, m-1$ **do**
2:     Define region $R_t = \left\{ x \in \mathbb{S}^{d-1} : \frac{b}{2} \leq w_t \cdot x \leq b \right\}$.
3:     Rejection sample $x_t \sim D_{\mathcal{X}}|_{R_t}$. In other words, draw $x_t$ from $D_{\mathcal{X}}$ until $x_t$ is in $R_t$. Query $\mathcal{O}$ for its label $y_t$.
4:     $w_{t+1} \leftarrow w_t - 2\mathbb{1}\{y_t w_t \cdot x_t < 0\} \cdot (w_t \cdot x_t) \cdot x_t$.
5: **end for**
6: **return** $w_m$.

## 5 Performance Guarantees

We show that ACTIVE-PERCEPTRON works in the bounded and the adversarial noise models, achieving computational efficiency and near-optimal label complexities. To this end, we first give a lower bound on the label complexity under bounded noise, and then give computational and label complexity upper bounds under the two noise conditions respectively. We defer all proofs to the Appendix.

### 5.1 A Lower Bound under Bounded Noise

We first present an information-theoretic lower bound on the label complexity in the bounded noise setting under uniform distribution. This extends the distribution-free lower bounds of [53, 37], and generalizes the realizable-case lower bound of [47] to the bounded noise setting. Our lower bound can also be viewed as an extension of [59]'s Theorem 3; specifically it addresses the hardness under the $\alpha$-Tsybakov noise condition where $\alpha = 0$ (while [59]'s Theorem 3 provides lower boundes when $\alpha \in (0, 1)$).

**Theorem 1.** *For any $d > 4$, $0 \leq \eta < \frac{1}{2}$, $0 < \epsilon \leq \frac{1}{4\pi}$, $0 < \delta \leq \frac{1}{4}$, for any active learning algorithm $\mathcal{A}$, there is a $u \in \mathbb{S}^{d-1}$, and a labeling oracle $\mathcal{O}$ that satisfies $\eta$-bounded noise condition with respect to $u$, such that if with probability at least $1 - \delta$, $\mathcal{A}$ makes at most $n$ queries of labels to $\mathcal{O}$ and outputs $v \in \mathbb{S}^{d-1}$ such that $\mathbb{P}[\text{sign}(v \cdot X) \neq \text{sign}(u \cdot X)] \leq \epsilon$, then $n \geq \Omega \left( \frac{d \log \frac{1}{\epsilon}}{(1-2\eta)^2} + \frac{\eta \log \frac{1}{\delta}}{(1-2\eta)^2} \right)$.*

### 5.2 Bounded Noise

We establish Theorem 2 in the bounded noise setting. The theorem implies that, with appropriate settings of input parameters, ACTIVE-PERCEPTRON efficiently learns a halfspace of excess error at most $\epsilon$ with probability at least $1 - \delta$, under the assumption that $D_{\mathcal{X}}$ is uniform over the unit sphere and $\mathcal{O}$ has bounded noise. In addition, it queries at most $\tilde{O}(\frac{d}{(1-2\eta)^2} \ln \frac{1}{\epsilon})$ labels. This matches the lower bound of Theorem 1, and improves over the state of the art result of [8], where a label complexity of $\tilde{O}(d^{O(\frac{1}{(1-2\eta)^4})} \ln \frac{1}{\epsilon})$ is shown using a different algorithm.

The proof and the precise setting of parameters ($m_k$ and $b_k$) are given in Appendix C.

**Theorem 2** (ACTIVE-PERCEPTRON under Bounded Noise)**.** *Suppose Algorithm 1 has inputs labeling oracle $\mathcal{O}$ that satisfies $\eta$-bounded noise condition with respect to halfspace $u$, initial halfspace $v_0$ such that $\theta(v_0, u) \in [0, \frac{\pi}{2}]$, target error $\epsilon$, confidence $\delta$, sample schedule $\{m_k\}$ where $m_k = \Theta \left( \frac{d}{(1-2\eta)^2} (\ln \frac{d}{(1-2\eta)^2} + \ln \frac{k}{\delta}) \right)$, band width $\{b_k\}$ where $b_k = \Theta \left( \frac{2^{-k}(1-2\eta)}{\sqrt{d} \ln(km_k/\delta)} \right)$. Then with probability at least $1 - \delta$:*

1. *The output halfspace $v$ is such that $\mathbb{P}[\text{sign}(v \cdot X) \neq \text{sign}(u \cdot X)] \leq \epsilon$.*

2. *The number of label queries is $O \left( \frac{d}{(1-2\eta)^2} \cdot \ln \frac{1}{\epsilon} \cdot \left( \ln \frac{d}{(1-2\eta)^2} + \ln \frac{1}{\delta} + \ln \ln \frac{1}{\epsilon} \right) \right)$.*

3. *The number of unlabeled examples drawn is*

$$O\left( \frac{d}{(1-2\eta)^3} \cdot \left( \ln \frac{d}{(1-2\eta)^2} + \ln \frac{1}{\delta} + \ln \ln \frac{1}{\epsilon} \right)^2 \cdot \frac{1}{\epsilon} \ln \frac{1}{\epsilon} \right).$$

4. *The algorithm runs in time* $O\left( \frac{d^2}{(1-2\eta)^3} \cdot \left( \ln \frac{d}{(1-2\eta)^2} + \ln \frac{1}{\delta} + \ln \ln \frac{1}{\epsilon} \right)^2 \cdot \frac{1}{\epsilon} \ln \frac{1}{\epsilon} \right).$

The theorem follows from Lemma 2 below. The key ingredient of the lemma is a delicate analysis of the dynamics of the angles $\{\theta_t\}_{t=0}^m$, where $\theta_t = \theta(w_t, u)$ is the angle between the iterate $w_t$ and the halfspace $u$. Since $x_t$ is randomly sampled and $y_t$ is noisy, we are only able to show that $\theta_t$ decreases by a decent amount *in expectation.* To remedy the stochastic fluctuations, we apply martingale concentration inequalities to carefully control the upper envelope of sequence $\{\theta_t\}_{t=0}^m$.

**Lemma 2** (MODIFIED-PERCEPTRON under Bounded Noise). *Suppose Algorithm 2 has inputs labeling oracle $\mathcal{O}$ that satisfies $\eta$-bounded noise condition with respect to halfspace $u$, initial halfspace $w_0$ and angle upper bound $\theta \in (0, \frac{\pi}{2}]$ such that $\theta(w_0, u) \leq \theta$, confidence $\delta$, number of iterations $m = \Theta(\frac{d}{(1-2\eta)^2}(\ln \frac{d}{(1-2\eta)^2} + \ln \frac{1}{\delta}))$, band width $b = \Theta\left( \frac{\theta(1-2\eta)}{\sqrt{d}\ln(m/\delta)} \right)$. Then with probability at least $1 - \delta$:*

1. *The output halfspace $w_m$ is such that $\theta(w_m, u) \leq \frac{\theta}{2}$.*

2. *The number of label queries is* $O\left( \frac{d}{(1-2\eta)^2} \left( \ln \frac{d}{(1-2\eta)^2} + \ln \frac{1}{\delta} \right) \right).$

3. *The number of unlabeled examples drawn is* $O\left( \frac{d}{(1-2\eta)^3} \cdot \left( \ln \frac{d}{(1-2\eta)^2} + \ln \frac{1}{\delta} \right)^2 \cdot \frac{1}{\theta} \right).$

4. *The algorithm runs in time* $O\left( \frac{d^2}{(1-2\eta)^3} \cdot \left( \ln \frac{d}{(1-2\eta)^2} + \ln \frac{1}{\delta} \right)^2 \cdot \frac{1}{\theta} \right).$

### 5.3 Adversarial Noise

We establish Theorem 3 in the adversarial noise setting. The theorem implies that, with appropriate settings of input parameters, ACTIVE-PERCEPTRON efficiently learns a halfspace of excess error at most $\epsilon$ with probability at least $1 - \delta$, under the assumption that $D_{\mathcal{X}}$ is uniform over the unit sphere and $\mathcal{O}$ has an adversarial noise of magnitude $\nu = \Omega(\frac{\epsilon}{\ln d + \ln \ln \frac{1}{\epsilon}})$. In addition, it queries at most $\tilde{O}(d \ln \frac{1}{\epsilon})$ labels. Our label complexity bound is information-theoretically optimal [47], and matches the state of the art result of [39]. The benefit of our approach is computational: it has a running time of $\tilde{O}(\frac{d^2}{\epsilon})$, while [39] needs to solve a convex optimization problem whose running time is some polynomial over $d$ and $\frac{1}{\epsilon}$ with an unspecified degree.

The proof and the precise setting of parameters ($m_k$ and $b_k$) are given in Appendix C.

**Theorem 3** (ACTIVE-PERCEPTRON under Adversarial Noise). *Suppose Algorithm 1 has inputs labeling oracle $\mathcal{O}$ that satisfies $\nu$-adversarial noise condition with respect to halfspace $u$, initial halfspace $v_0$ such that $\theta(v_0, u) \leq \frac{\pi}{2}$, target error $\epsilon$, confidence $\delta$, sample schedule $\{m_k\}$ where $m_k = \Theta(d(\ln d + \ln \frac{k}{\delta}))$, band width $\{b_k\}$ where $b_k = \Theta\left( \frac{2^{-k}}{\sqrt{d}\ln(km_k/\delta)} \right)$. Additionally $\nu \leq \Omega(\frac{\epsilon}{\ln \frac{d}{\delta} + \ln \ln \frac{1}{\epsilon}})$. Then with probability at least $1 - \delta$:*

1. *The output halfspace $v$ is such that $\mathbb{P}[\text{sign}(v \cdot X) \neq \text{sign}(u \cdot X)] \leq \epsilon$.*

2. *The number of label queries is* $O\left( d \cdot \ln \frac{1}{\epsilon} \cdot \left( \ln d + \ln \frac{1}{\delta} + \ln \ln \frac{1}{\epsilon} \right) \right).$

3. *The number of unlabeled examples drawn is* $O\left( d \cdot \left( \ln d + \ln \frac{1}{\delta} + \ln \ln \frac{1}{\epsilon} \right)^2 \cdot \frac{1}{\epsilon} \ln \frac{1}{\epsilon} \right).$

4. *The algorithm runs in time* $O\left( d^2 \cdot \left( \ln d + \ln \frac{1}{\delta} + \ln \ln \frac{1}{\epsilon} \right)^2 \cdot \frac{1}{\epsilon} \ln \frac{1}{\epsilon} \right).$

The theorem follows from Lemma 3 below, whose proof is similar to Lemma 2.

**Lemma 3** (MODIFIED-PERCEPTRON under Adversarial Noise). *Suppose Algorithm 2 has inputs labeling oracle $\mathcal{O}$ that satisfies $\nu$-adversarial noise condition with respect to halfspace $u$, initial halfspace $w_0$ and angle upper bound $\theta \in (0, \frac{\pi}{2}]$ such that $\theta(w_0, u) \leq \theta$, confidence $\delta$, number of iterations $m = \Theta(d(\ln d + \ln \frac{1}{\delta}))$, band width $b = \Theta\left(\frac{\theta}{\sqrt{d}\ln(m/\delta)}\right)$. Additionally $\nu \leq \Omega(\frac{\theta}{\ln(m/\delta)})$. Then with probability at least $1 - \delta$:*

1. *The output halfspace $w_m$ is such that $\theta(w_m, u) \leq \frac{\theta}{2}$.*

2. *The number of label queries is $O\left(d \cdot \left(\ln d + \ln \frac{1}{\delta}\right)\right)$.*

3. *The number of unlabeled examples drawn is $O\left(d \cdot \left(\ln d + \ln \frac{1}{\delta}\right)^2 \cdot \frac{1}{\theta}\right)$*

4. *The algorithm runs in time $O\left(d^2 \cdot \left(\ln d + \ln \frac{1}{\delta}\right)^2 \cdot \frac{1}{\theta}\right)$.*

# 6   Implications to Passive Learning

ACTIVE-PERCEPTRON can be converted to a passive learning algorithm, PASSIVE-PERCEPTRON, for learning homogeneous halfspaces under the uniform distribution over the unit sphere. PASSIVE-PERCEPTRON has PAC sample complexities close to the lower bounds under the two noise models. We give a formal description of PASSIVE-PERCEPTRON in Appendix B. We give its formal guarantees in the corollaries below, which are immediate consequences of Theorems 2 and 3.

In the $\eta$-bounded noise model, the sample complexity of PASSIVE-PERCEPTRON improves over the state of the art result of [8], where a sample complexity of $\tilde{O}(\frac{d^{O(\frac{1}{(1-2\eta)^4})}}{\epsilon})$ is obtained. The bound has the same dependency on $\epsilon$ and $d$ as the minimax upper bound of $\tilde{\Theta}(\frac{d}{\epsilon(1-2\eta)})$ by [49], which is achieved by a computationally inefficient ERM algorithm.

**Corollary 1** (PASSIVE-PERCEPTRON under Bounded Noise). *Suppose PASSIVE-PERCEPTRON has inputs distribution $D$ that satisfies $\eta$-bounded noise condition with respect to $u$, initial halfspace $v_0$, target error $\epsilon$, confidence $\delta$, sample schedule $\{m_k\}$ where $m_k = \Theta\left(\frac{d}{(1-2\eta)^2}(\ln\frac{d}{(1-2\eta)^2} + \ln\frac{k}{\delta})\right)$, band width $\{b_k\}$ where $b_k = \Theta\left(\frac{2^{-k}(1-2\eta)}{\sqrt{d}\ln(km_k/\delta)}\right)$. Then with probability at least $1 - \delta$: (1) The output halfspace $v$ is such that $\mathrm{err}(h_v) \leq \mathrm{err}(h_u) + \epsilon$; (2) The number of labeled examples drawn is $\tilde{O}\left(\frac{d}{(1-2\eta)^3\epsilon}\right)$. (3) The algorithm runs in time $\tilde{O}\left(\frac{d^2}{(1-2\eta)^3\epsilon}\right)$.*

In the $\nu$-adversarial noise model, the sample complexity of PASSIVE-PERCEPTRON matches the minimax optimal sample complexity upper bound of $\tilde{\Theta}(\frac{d}{\epsilon})$ obtained in [39]. Same as in active learning, our algorithm has a faster running time than [39].

**Corollary 2** (PASSIVE-PERCEPTRON under Adversarial Noise). *Suppose PASSIVE-PERCEPTRON has inputs distribution $D$ that satisfies $\nu$-adversarial noise condition with respect to $u$, initial halfspace $v_0$, target error $\epsilon$, confidence $\delta$, sample schedule $\{m_k\}$ where $m_k = \Theta\left(d(\ln d + \ln\frac{k}{\delta})\right)$, band width $\{b_k\}$ where $b_k = \Theta\left(\frac{2^{-k}}{\sqrt{d}\ln(km_k/\delta)}\right)$. Furthermore $\nu = \Omega(\frac{\epsilon}{\ln\ln\frac{1}{\epsilon}+\ln\frac{d}{\delta}})$. Then with probability at least $1 - \delta$: (1) The output halfspace $v$ is such that $\mathrm{err}(h_v) \leq \mathrm{err}(h_u) + \epsilon$; (2) The number of labeled examples drawn is $\tilde{O}\left(\frac{d}{\epsilon}\right)$. (3) The algorithm runs in time $\tilde{O}\left(\frac{d^2}{\epsilon}\right)$.*

Tables 3 and 4 present comparisons between our results and results most closely related to ours.

**Acknowledgments.** The authors thank Kamalika Chaudhuri for help and support, Hongyang Zhang for thought-provoking initial conversations, Jiapeng Zhang for helpful discussions, and the anonymous reviewers for their insightful feedback. Much of this work is supported by NSF IIS-1167157 and 1162581.

Table 3: A comparison of algorithms for PAC learning halfspaces under the uniform distribution, in the $\eta$-bounded noise model.

| Algorithm | Sample Complexity | Time Complexity |
|---|---|---|
| [8] | $\tilde{O}(\frac{d^{O(\frac{1}{(1-2\eta)^4})}}{\epsilon})$ | $\tilde{O}(\frac{d^{O(\frac{1}{(1-2\eta)^4})}}{\epsilon})$ |
| ERM [49] | $\tilde{O}(\frac{d}{(1-2\eta)\epsilon})$ | superpoly$(d, \frac{1}{\epsilon})$ |
| Our Work | $\tilde{O}(\frac{d}{(1-2\eta)^3\epsilon})$ | $\tilde{O}(\frac{d^2}{(1-2\eta)^3} \cdot \frac{1}{\epsilon})$ |

Table 4: A comparison of algorithms for PAC learning halfspaces under the uniform distribution, in the $\nu$-adversarial noise model where $\nu = \Omega(\frac{\epsilon}{\ln\ln\frac{1}{\epsilon}+\ln d})$.

| Algorithm | Sample Complexity | Time Complexity |
|---|---|---|
| [39] | $\tilde{O}(\frac{d}{\epsilon})$ | poly$(d, \frac{1}{\epsilon})$ |
| ERM [57] | $\tilde{O}(\frac{d}{\epsilon})$ | superpoly$(d, \frac{1}{\epsilon})$ |
| Our Work | $\tilde{O}(\frac{d}{\epsilon})$ | $\tilde{O}(\frac{d^2}{\epsilon})$ |

## Footnotes

[3] Note that the adversarial noise model is not the same as that in online learning [18], where each example can be chosen adversarially.

[4] The label complexity bound is implicit in [39] by a refined analysis of the algorithm of [6] (See their Lemma 8 for details).

[5]The algorithm needs to minimize 0-1 loss, the best known method for which requires superpolynomial time.

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
