[Supplementary Material · al_linsep_supp.pdf]

# A    Additional Related Work

**Active Learning.**    The recent decades have seen much success in both theory and practice of active learning; see the excellent surveys by [54, 37, 25]. On the theory side, many label-efficient active learning algorithms have been proposed and analyzed [21, 31, 24, 11, 34, 10, 27, 14, 16, 35, 46, 40, 15, 58, 36, 2, 60, 41]. Most algorithms are *disagreement-based* algorithms [37], and are not label-optimal due to the conservativeness of their label query policy. In addition, most of these algorithms require either explicit enumeration of classifiers in the hypothesis classes, or solving empirical 0-1 loss minimization problems on sets of examples. The former approach is easily seen to be computationally infeasible, while the latter is proven to be computationally hard as well [5]. The only exception in this family we are aware of is [38]. [38] considers active learning by sequential convex surrogate loss minimization. However, it assumes that the expected convex loss minimizer over all possible functions lies in a pre-specified real-valued function class, which is unlikely to hold in the bounded noise and the adversarial noise settings.

Some recent works [60, 41, 10, 9, 59] provide noise-tolerant active learning algorithms with improved label complexity over disagreement-based approaches. However, they are still computationally inefficient: [60] relies on solving a series of linear program with an exponential number of constraints, which are computationally intractable; [41, 10, 9, 59] relies on solving a series of empirical 0-1 loss minimization problems, which are also computationally hard in the presence of noise [5].

**Efficient Learning of Halfspaces.**    A series of papers have shown the hardness of learning halfspaces with agnostic noise [5, 30, 33, 44, 23]. These results indicate that, to have nontrivial guarantees on learning halfspaces with noise in polynomial time, one has to make additional assumptions on the data distribution over instances and labels.

Many noise models, other than the bounded noise model and the adversarial noise model, has been studied in the literature. A line of work [19, 52, 28, 1] considers parameterized noise models. For instance, [28] gives an efficient algorithm for the setting that $\mathbb{E}[Y|X = x] = u \cdot x$ where $u$ is the optimal classifier. [1] studies a generalization of the above linear noise model, where $Y$ is a multiclass label, and there is a link function $\Phi$ such that $\mathbb{E}[Y|X = x] = \nabla \Phi(u \cdot x)$. Their analyses depend heavily on the noise models and it is unknown whether their algorithms can work with more general noise settings. [61] analyzes the problem of learning halfspaces under a new noise condition (as an application of their general analysis of stochastic gradient Langevin dynamics). They assume that the label flipping probability on every $x$ is bounded by $\frac{1}{2} - c|u \cdot x|$, for some $c \in (0, \frac{1}{2}]$. It can be seen that the bounded noise condition implies the noise condition of [61], and it is an interesting open question whether it is possible to extend our algorithm and analysis to their setting.

Under the random classification noise condition [3], [17] gives the first efficient passive learning algorithm of learning halfspaces, by using a modification of Perceptron update (similar to Equation (1)) together with a boosting-type aggregation. [12] proposes an active statistical query algorithm for learning halfspaces. The algorithm proceeds by estimating the distance between the current halfspace and the optimal halfspace. However, it requires a suboptimal number of $\tilde{O}(\frac{d^2}{(1-2\eta)^2})$ labels. In addition, both results above rely on the uniformity over the random classification noise, and it is shown in [7] that this type of statistical query algorithms will fail in the heterogeneous noise setting (in particular the bounded noise setting and the adversarial noise setting).

In the adversarial noise model, we assume that there is a halfspace $u$ with error at most $\nu$ over data. The goal is to design an efficient algorithm that outputting a classifier that disagrees with $u$ with probability at most $\epsilon$. [42] proposes an elegant averaging-based algorithm that tolerates an error of at most $\nu = \Omega(\frac{\epsilon}{\ln \frac{1}{\epsilon}})$ assuming that the unlabeled distribution is uniform. However it has a suboptimal label complexity of $\tilde{O}(\frac{d^2}{\epsilon^2})$. Under the assumption that the unlabeled distribution is log-concave or $s$-concave, the state of the art results [6, 13] give efficient margin-based algorithms that tolerates a noise of $\nu = \tilde{\Omega}(\epsilon)$. As discussed in the main text, such algorithms require a hinge loss minimization procedure that has a running time polynomial in $d$ with an unspecified degree. Finally, [23] gives a PTAS that outputs a classifier with error $(1 + \mu)\nu + \epsilon$, in time $O(\text{poly}(d^{\tilde{O}(\frac{1}{\mu^2})}, \frac{1}{\epsilon}))$. Observe that in the case of $\nu = O(\epsilon)$, the running time is an unspecified high order polynomial in terms of $d$ and $\frac{1}{\epsilon}$.

# B   Implications to Passive Learning

In this section, we formally describe PASSIVE-PERCEPTRON (Algorithm 3), a passive learning version of Algorithm 1. The algorithmic framework is similar to Algorithm 1, except that it calls Algorithm 4 rather than Algorithm 2.

---

**Algorithm 3** PASSIVE-PERCEPTRON

---

**Input:** Initial halfspace $v_0$, target error $\epsilon$, confidence $\delta$, sample schedule $\{m_k\}$, band width $\{b_k\}$.
**Output:** learned halfspace $\hat{v}$.
  1: Let $k_0 = \lceil \log_2 \frac{1}{\epsilon} \rceil$.
  2: **for** $k = 1, 2, \ldots, k_0$ **do**
  3:     $v_k \leftarrow$ PASSIVE-MODIFIED-PERCEPTRON$(\mathcal{O}, v_{k-1}, \frac{\pi}{2^k}, \frac{\delta}{k(k+1)}, m_k, b_k)$.
  4: **end for**
  5: **return** $v_{k_0}$.

---

Algorithm 4 is similar to Algorithm 2, except that it draws labeled examples from $D$ directly, as opposed to performing label queries on unlabeled examples drawn.

---

**Algorithm 4** PASSIVE-MODIFIED-PERCEPTRON

---

**Input:** Initial halfspace $w_0$, angle upper bound $\theta$, confidence $\delta$, number of iterations $m$, band width $b$.
**Output:** Improved halfspace $w_m$.
  1: **for** $t = 0, 1, 2, \ldots, m-1$ **do**
  2:     Define region $C_t = \left\{ (x, y) \in \mathbb{S}^{d-1} \times \{-1, +1\} : \frac{b}{2} \leq w_t \cdot x \leq b \right\}$.
  3:     Rejection sample $(x_t, y_t) \sim D|_{C_t}$. In other words, repeat drawing example $(x_t, y_t) \sim D$ until it is in $C_t$.
  4:     $w_{t+1} \leftarrow w_t - 2\mathbb{1}\{y_t w_t \cdot x_t < 0\} \cdot (w_t \cdot x_t) \cdot x_t$.
  5: **end for**
  6: **return** $w_m$.

---

It can be seen that with the same input as ACTIVE-PERCEPTRON, PASSIVE-PERCEPTRON has exactly the same running time, and the number of labeled examples drawn in PASSIVE-PERCEPTRON is exactly the same as the number of unlabeled examples drawn in ACTIVE-PERCEPTRON. Therefore, Corollaries 1 and 2 are immediate consequences of Theorems 2 and 3.

# C   Proofs of Theorems 2 and 3

In this section, we give straightforward proofs that show Theorem 2 (resp. Theorem 3) are direct consequences of Lemma 2 (resp. Lemma 3). We defer the proofs of Lemmas 2 and 3 to Appendix D.

**Theorem 4** (Theorem 2 Restated)**.** *Suppose Algorithm 1 has inputs labeling oracle $\mathcal{O}$ that satisfies $\eta$-bounded noise condition with respect to underlying halfspace $u$, initial halfspace $v_0$ such that $\theta(v_0, u) \leq \frac{\pi}{2}$, target error $\epsilon$, confidence $\delta$, sample schedule $\{m_k\}$ where $m_k = \lceil \frac{(3200\pi)^3 d}{(1-2\eta)^2} (\ln \frac{(3200\pi)^3 d}{(1-2\eta)^2} + \ln \frac{k(k+1)}{\delta}) \rceil$, band width $\{b_k\}$ where $b_k = \frac{1}{2(600\pi)^2 \ln \frac{m_k^2 k(k+1)}{\delta}} \frac{2^{-k}\pi(1-2\eta)}{\sqrt{d}}$.*
*Then with probability at least $1 - \delta$:*

    *1. The output halfspace $v$ is such that $\mathbb{P}[\text{sign}(v \cdot X) \neq \text{sign}(u \cdot X)] \leq \epsilon$.*

    *2. The number of label queries is $O\left( \frac{d}{(1-2\eta)^2} \cdot \ln \frac{1}{\epsilon} \cdot \left( \ln \frac{d}{(1-2\eta)^2} + \ln \frac{1}{\delta} + \ln \ln \frac{1}{\epsilon} \right) \right)$.*

    *3. The number of unlabeled examples drawn is $O\left( \frac{d}{(1-2\eta)^3} \cdot \left( \ln \frac{d}{(1-2\eta)^2} + \ln \frac{1}{\delta} + \ln \ln \frac{1}{\epsilon} \right)^2 \cdot \frac{1}{\epsilon} \ln \frac{1}{\epsilon} \right)$.*

    *4. The algorithm runs in time $O\left( \frac{d^2}{(1-2\eta)^3} \cdot \left( \ln \frac{d}{(1-2\eta)^2} + \ln \frac{1}{\delta} + \ln \ln \frac{1}{\epsilon} \right)^2 \cdot \frac{1}{\epsilon} \ln \frac{1}{\epsilon} \right)$.*

*Proof of Theorem 4.* From Lemma 2, we know that for every $k$, there is an event $E_k$ such that $\mathbb{P}(E_k) \geq 1 - \frac{\delta}{k(k+1)}$, and on event $E_k$, items 1 to 4 of Lemma 2 hold for input $w_0 = v_{k-1}$, output $w_m = v_k$, $\theta = \frac{\pi}{2^k}$, $\delta = \frac{\delta}{k(k+1)}$.

Define event $E = \cup_{k=1}^{k_0} E_k$. By union bound, $\mathbb{P}(E) \geq 1 - \delta$. We henceforth condition on event $E$ happening.

1. By induction, the final output $v = v_{k_0}$ is such that $\theta(v, u) \leq 2^{-k_0}\pi \leq \epsilon\pi$, implying that $\mathbb{P}[\text{sign}(v \cdot X) \neq \text{sign}(u \cdot X)] \leq \epsilon$.

2. Define the number of label queries to oracle $\mathcal{O}$ at iteration $k$ as $m_k$. On event $E_k$, $m_k$ is at most $O\left( \frac{d}{(1-2\eta)^2} \left( \ln \frac{d}{(1-2\eta)^2} + \ln \frac{k}{\delta} \right) \right)$. Thus, the total number of label queries to oracle $\mathcal{O}$ is $\sum_{k=1}^{k_0} m_k$, which is at most

$$k_0 \cdot m_{k_0} = O\left( k_0 \cdot \frac{d}{(1-2\eta)^2} \left( \ln \frac{d}{(1-2\eta)^2} + \ln \frac{k_0}{\delta} \right) \right).$$

Item 2 is proved by noting that $k_0 \leq \log \frac{1}{\epsilon} + 1$.

3. Define the number of unlabeled examples drawn iteration $k$ as $n_k$. On event $E_k$, $n_k$ is at most $O\left( \frac{d}{(1-2\eta)^3} \cdot \left( \ln \frac{d}{(1-2\eta)^2} + \ln \frac{k}{\delta} \right)^2 \cdot \frac{1}{\epsilon} \right)$. Thus, the total number of unlabeled examples drawn is $\sum_{k=1}^{k_0} n_k$, which is at most

$$k_0 n_{k_0} = O\left( k_0 \cdot \frac{d}{(1-2\eta)^3} \cdot \left( \ln \frac{d}{(1-2\eta)^2} + \ln \frac{k_0}{\delta} \right)^2 \cdot \frac{1}{\epsilon} \right).$$

Item 3 is proved by noting that $k_0 \leq \log \frac{1}{\epsilon} + 1$.

4. Item 4 is immediate from Item 3 and the fact that the time for processing each example is at most $O(d)$. $\qquad\square$

**Theorem 5** (Theorem 3 Restated). *Suppose Algorithm 1 has inputs labeling oracle $\mathcal{O}$ that satisfies $\nu$-adversarial noise condition with respect to underlying halfspace $u$, initial halfspace $v_0$ such that $\theta(v_0, u) \leq \frac{\pi}{2}$, target error $\epsilon$, confidence $\delta$, sample schedule $\{m_k\}$ where $m_k = \lceil (3200\pi)^3 d(\ln(3200\pi)^3 d + \ln \frac{k(k+1)}{\delta}) \rceil$, band width $\{b_k\}$ where $b_k = \frac{1}{2(600\pi)^2 \ln \frac{m_k^2 k(k+1)}{\delta}} \frac{2^{-k}\pi}{\sqrt{d}}$. Additionally $\nu \leq \frac{\epsilon}{384(600\pi)^4(4\ln((3200\pi)^3 d)+8\ln\ln\frac{1}{\epsilon}+\ln\frac{1}{\delta})}$. Then with probability at least $1 - \delta$:*

1. *The output halfspace $v$ is such that $\mathbb{P}[\text{sign}(v \cdot X) \neq \text{sign}(u \cdot X)] \leq \epsilon$.*

2. *The number of label queries is $O\left( d \cdot \ln \frac{1}{\epsilon} \cdot \left( \ln d + \ln \frac{1}{\delta} + \ln\ln \frac{1}{\epsilon} \right) \right)$.*

3. *The number of unlabeled examples drawn is $O\left( d \cdot \left( \ln d + \ln \frac{1}{\delta} + \ln\ln \frac{1}{\epsilon} \right)^2 \cdot \frac{1}{\epsilon} \ln \frac{1}{\epsilon} \right)$.*

4. *The algorithm runs in time $O\left( d^2 \cdot \left( \ln d + \ln \frac{1}{\delta} + \ln\ln \frac{1}{\epsilon} \right)^2 \cdot \frac{1}{\epsilon} \ln \frac{1}{\epsilon} \right)$.*

*Proof of Theorem 5.* From Lemma 3, we know that for every $k$, there is an event $E_k$ such that $\mathbb{P}(E_k) \geq 1 - \frac{\delta}{k(k+1)}$, and on event $E_k$, items 1 to 4 of Lemma 3 hold for input $w_0 = v_k$, output $w_m = v_{k+1}$, $\theta = \frac{\pi}{2^k}$.

Define event $E = \cup_{k=1}^{k_0} E_k$. By union bound, $\mathbb{P}(E) \geq 1 - \delta$. We henceforth condition on event $E$ happening.

1. By induction, the final output $v = v_{k_0}$ is such that that $\theta(v, u) \leq 2^{-k_0}\pi \leq \epsilon\pi$, implying that $\mathbb{P}[\text{sign}(v \cdot X) \neq \text{sign}(u \cdot X)] \leq \epsilon$.

2. Define the number of label queries to oracle $\mathcal{O}$ at iteration $k$ as $m_k$. On event $E_k$, $m_k$ is at most $O\left(d\left(\ln d + \ln \frac{k}{\delta}\right)\right)$. Thus, the total number of label queries to oracle $\mathcal{O}$ is $\sum_{k=1}^{k_0} m_k$, which is at most

$$k_0 \cdot m_{k_0} = O\left(k_0 \cdot d \left(\ln d + \ln \frac{k_0}{\delta}\right)\right).$$

Item 2 is proved by noting that $k_0 \le \log \frac{1}{\epsilon} + 1$.

3. Define the number of unlabeled examples drawn iteration $k$ as $n_k$. On event $E_k$, $n_k$ is at most $O\left(d \cdot \left(\ln d + \ln \frac{k}{\delta}\right)^2 \cdot \frac{1}{\epsilon}\right)$. Thus, the total number of unlabeled examples drawn is $\sum_{k=1}^{k_0} n_k$, which is at most

$$k_0 n_{k_0} = O\left(k_0 \cdot d \cdot \left(\ln d + \ln \frac{k_0}{\delta}\right)^2 \cdot \frac{1}{\epsilon}\right).$$

Item 3 is proved by noting that $k_0 \le \log \frac{1}{\epsilon} + 1$.

4. Item 4 is immediate from Item 3 and the fact that the time for processing each example is at most $O(d)$. $\qquad\square$

## D   Performance Guarantees of MODIFIED-PERCEPTRON

In this section, we prove Lemmas 2 and 3, which guarantees the shrinkage of $\theta_t$. Two major building blocks of Lemma 2 (resp. Lemma 3) are Lemmas 7 and 9 (resp. Lemmas 7 and 10). In essence, Lemma 7 turns per-iteration in-expectation guarantees provided by Lemmas 9 and 10 into high probability upper bounds on the final $\theta_m$. We present Lemma 7 and its proof in detail in this section, and defer Lemmas 9 and 10 to Appendix E.

**Lemma 4** (Lemma 2 Restated). *Suppose Algorithm 2 has inputs labeling oracle $\mathcal{O}$ that satisfies $\eta$-bounded noise condition with respect to underlying halfspace $u$, initial vector $w_0$ and angle upper bound $\theta \in (0, \frac{\pi}{2})$ such that $\theta(w_0, u) \le \theta$, confidence $\delta$, number of iterations $m = \lceil \frac{(3200\pi)^3 d}{(1-2\eta)^2} (\ln \frac{(3200\pi)^3 d}{(1-2\eta)^2} + \ln \frac{1}{\delta}) \rceil$, band width $b = \frac{1}{2(600\pi)^2 \ln \frac{m^2}{\delta}} \frac{\theta(1-2\eta)}{\sqrt{d}}$. then with probability at least $1 - \delta$:*

1. *The output halfspace $w_m$ is such that $\theta(w_m, u) \le \frac{\theta}{2}$.*

2. *The number of label queries is $O\left(\frac{d}{(1-2\eta)^2}\left(\ln \frac{d}{(1-2\eta)^2} + \ln \frac{1}{\delta}\right)\right)$.*

3. *The number of unlabeled examples drawn is $O\left(\frac{d}{(1-2\eta)^3} \cdot \left(\ln \frac{d}{(1-2\eta)^2} + \ln \frac{1}{\delta}\right)^2 \cdot \frac{1}{\theta}\right)$.*

4. *The algorithm runs in time $O\left(\frac{d^2}{(1-2\eta)^3} \cdot \left(\ln \frac{d}{(1-2\eta)^2} + \ln \frac{1}{\delta}\right)^2 \cdot \frac{1}{\theta}\right)$.*

*Proof of Lemma 4.* We show that each item holds with high probability respectively.

1. It can be verified that conditions for Lemma 7 are satisfied with $\zeta = 1 - 2\eta$ (item 3 in the condition follows from Lemma 9, and item 4 in the condition follows from Lemma 6). This shows that items 1 with probability at least $1 - \delta/2$.

2. By the definition of $m$, the number of label queries is $m = O\left(\frac{d}{(1-2\eta)^2} \log \frac{d}{\delta(1-2\eta)^2}\right)$.

3. As for the number of unlabeled examples drawn by the algorithm, at each iteration $t \in [0, m]$, it takes $Z_t$ trials to hit an example in $[\frac{b}{2}, b]$, where $Z_t$ is a $\mathrm{Geometric}(p)$ random variable with $p = \mathbb{P}_{x \sim D_{\mathcal{X}}}[w_t \cdot x \in [\frac{b}{2}, b]]$. From Lemma 18, $p \geq \frac{\sqrt{d}}{8\pi} b = \frac{\tilde{c}(1-2\eta)\theta}{8\pi} = \Omega(\frac{(1-2\eta)\theta}{\ln \frac{d}{\delta(1-2\eta)^2}})$.

Define event

$$E := \left\{ Z_1 + \ldots + Z_m \leq \frac{2m}{p} \right\}$$

From Lemma 16 and the choice of $m$, $\mathbb{P}[E] \geq 1 - \frac{\delta}{2}$. Thus, on event $E$, the total number of unlabeled examples drawn is at most $\frac{2m}{p} = O(\frac{d}{(1-2\eta)^3} \log^2 \frac{d}{\delta(1-2\eta)^2} \frac{1}{\theta})$.

4. Observe that the time complexity for processing each example is at most $O(d)$. This shows that on event $E$, the total running time of the algorithm is at most $O(d \cdot \frac{2m}{p}) = O(\frac{d^2}{(1-2\eta)^3} \log^2 \frac{d}{\delta(1-2\eta)^2} \frac{1}{\theta})$.

Therefore, by a union bound, with probability at least $1 - \delta$, items 1 to 4 hold simultaneously. $\qquad \square$

**Lemma 5** (Lemma 3 restated). *Suppose Algorithm 2 has inputs labeling oracle $\mathcal{O}$ that satisfies $\nu$-adversarial noise condition with respect to underlying halfspace $u$, initial vector $w_0$ and angle upper bound $\theta$ such that $\theta(w_0, u) \leq \theta$, confidence $\delta$, number of iterations $m = \lceil(3200\pi)^3 d \ln \frac{(3200\pi)^3 d}{\delta}\rceil$, band width $b = \frac{1}{2(600\pi)^2 \ln \frac{m^2}{\delta}} \cdot \frac{\theta}{\sqrt{d}}$. Additionally $\nu \leq \frac{\theta}{384(600\pi)^4 \ln \frac{m^2}{\delta}}$. Then with probability at least $1 - \delta$:*

1. *The output halfspace $w_m$ is such that $\theta(w_m, u) \leq \frac{\theta}{2}$.*

2. *The number of label queries is $O\left(d \cdot \left(\ln d + \ln \frac{1}{\delta}\right)\right)$.*

3. *The number of unlabeled examples drawn is $O\left(d \cdot \left(\ln d + \ln \frac{1}{\delta}\right)^2 \cdot \frac{1}{\theta}\right)$*

4. *The algorithm runs in time $O\left(d^2 \cdot \left(\ln d + \ln \frac{1}{\delta}\right)^2 \cdot \frac{1}{\theta}\right)$.*

*Proof of Lemma 5.* We show that each item holds with high probability respectively.

1. It can be verified that conditions for Lemma 7 are satisfied with $\zeta = 1$ (item 3 in the condition follows from Lemma 10, and item 4 in the condition follows from Lemma 6). This gives items 1 with probability at least $1 - \delta/2$.

2. By the definition of $m$, the number of label queries is $m = O\left(d \cdot \left(\ln d + \ln \frac{1}{\delta}\right)\right)$.

3. The number of unlabeled examples drawn by the algorithm can be analyzed similarly as in the previous proof, which is at most $\frac{2m}{p} = O\left(d \cdot \left(\ln d + \ln \frac{1}{\delta}\right)^2 \cdot \frac{1}{\theta}\right)$ with probability at least $1 - \delta/2$.

4. Observe that the time complexity for processing each example is at most $O(d)$. This gives that on event $E$, the total running time of the algorithm is at most $O(d \cdot \frac{2m}{p}) = O\left(d^2 \cdot \left(\ln d + \ln \frac{1}{\delta}\right)^2 \cdot \frac{1}{\theta}\right)$.

Therefore, by a union bound, with probability at least $1 - \delta$, items 1 to 4 hold simultaneously. $\qquad \square$

Next we show a technical lemma used in the above proofs, coarsely bounding the difference between $\cos \theta_{t+1}$ and $\cos \theta_t$.

**Lemma 6.** *Suppose* $0 < \tilde{c}, \zeta < 1$, $b = \frac{\tilde{c}\zeta\theta}{\sqrt{d}} \leq 1$, *and* $(x_t, y_t)$ *is drawn from distribution* $D|_{R_t}$ *where* $R_t = \left\{(x, y) : x \cdot w_t \in [\frac{b}{2}, b]\right\}$. *If unit vector* $w_t$ *has angle* $\theta_t$ *with* $u$ *such that* $\theta_t \leq \frac{5}{3}\theta$, *then update* (5) *has the following guarantee:* $|\cos\theta_{t+1} - \cos\theta_t| \leq \frac{16\tilde{c}\zeta\theta^2}{3\sqrt{d}}$.

*Proof.* By Lemma 8,

$$\cos\theta_{t+1} - \cos\theta_t = -2\mathbb{1}\left\{y_t \neq \text{sign}(w_t \cdot x_t)\right\}(w_t \cdot x_t) \cdot (u \cdot x_t).$$

Firstly, note $|\cos\theta_{t+1} - \cos\theta_t| \leq 2|w_t \cdot x_t||u \cdot x_t| \leq 2b|u \cdot x_t|$.

Observe that

$$
\begin{aligned}
&|u \cdot x_t| \\
\leq\ & |w_t \cdot x_t| + |(u - w_t) \cdot x_t| \\
\leq\ & b + 2\sin\frac{\theta_t}{2} \\
\leq\ & b + \theta_t
\end{aligned}
$$

Thus, we have $|\cos\theta_{t+1} - \cos\theta_t| \leq 2b(b + \theta_t) = \frac{2\tilde{c}^2\zeta^2\theta^2}{d} + \frac{2\tilde{c}\zeta\theta\theta_t}{\sqrt{d}} \leq \frac{16\tilde{c}\zeta\theta^2}{3\sqrt{d}}$. $\qquad\square$

**Lemma 7.** *Suppose* $0 < \zeta < 1$, *and the following conditions hold:*

1. *Initial unit vector* $w_0$ *has angle* $\theta_0 = \theta(w_0, u) \leq \theta \leq \frac{27}{50}\pi$ *with* $u$;

2. *Integer* $m = \lceil \frac{(3200\pi)^3 d}{\zeta^2}(\ln\frac{(3200\pi)^3 d}{\zeta^2} + \ln\frac{1}{\delta}) \rceil$ *and* $\tilde{c} = \frac{1}{2(600\pi)^2 \ln\frac{m^2}{\delta}}$;

3. *For all* $t$, *if* $\frac{1}{4}\theta \leq \theta_t \leq \frac{5}{3}\theta$, *then* $\mathbb{E}[\cos\theta_{t+1} - \cos\theta_t|\theta_t] \geq \frac{\tilde{c}}{100\pi}\frac{\zeta^2\theta^2}{d}$;

4. *For all* $t$, *if* $\theta_t \leq \frac{5}{3}\theta$, *then* $|\cos\theta_{t+1} - \cos\theta_t| \leq \frac{16\tilde{c}\zeta\theta^2}{3\sqrt{d}}$ *holds with probability 1.*

*Then with probability at least* $1 - \delta/2$, $\theta_m \leq \frac{1}{2}\theta$.

*Proof.* Define random variable $D_t$ as:

$$D_t := \left(\cos\theta_{t+1} - \cos\theta_t - \frac{\tilde{c}}{100\pi}\frac{\zeta^2\theta^2}{d}\right)\mathbb{1}\left\{\frac{1}{4}\theta \leq \theta_t \leq \frac{5}{3}\theta\right\}$$

Note that $\mathbb{E}[D_t|\theta_t] \geq 0$ and from Lemma 6, $|D_t| \leq |\cos\theta_{t+1} - \cos\theta_t| + \frac{\tilde{c}}{100\pi}\frac{\zeta^2\theta^2}{d} \leq \frac{6\tilde{c}\zeta\theta^2}{\sqrt{d}}$. Therefore, $\{D_t\}$ is a bounded submartingale difference sequence. By Azuma's Inequality (see Lemma 15) and union bound, define event

$$E = \left\{\text{for all } 0 \leq t_1 \leq t_2 \leq m, \sum_{s=t_1}^{t_2-1} D_s \geq -\frac{6\tilde{c}\zeta\theta^2}{\sqrt{d}}\sqrt{2(t_2 - t_1)\ln\frac{2m^2}{\delta}}\right\}$$

Then $\mathbb{P}(E) \geq 1 - \frac{\delta}{2}$.

We now condition on event $E$. We break the subsequent analysis into two parts: (1) Show that there exists some $t$ such that $\theta_t$ goes below $\frac{1}{4}\theta$. (2) Show that $\theta_t$ must stay below $\frac{1}{2}\theta$ afterwards.

1. First, it can be checked by algebra that $m \geq \frac{200\pi d}{\zeta^2\tilde{c}}$. We show the following claim.

   **Claim 1.** *There exists some* $t \in [0, m]$, *such that* $\theta_t < \frac{1}{4}\theta$.

*Proof.* We first show that it is impossible for all $t \in [0, m]$ such that $\theta_t \in \left[\frac{1}{4}\theta, \frac{5}{3}\theta\right]$. To this end, assume this holds for the sake of contradiction. In this case, for all $t \in [0, m]$, $D_t = \cos\theta_{t+1} - \cos\theta_t - \frac{\tilde{c}}{100\pi}\frac{\zeta^2\theta^2}{d}$. Therefore,

$$\cos\theta_m - \cos\theta_0$$

$$= \sum_{s=0}^{m-1} D_s + \frac{\tilde{c}}{100\pi}\frac{\zeta^2\theta^2}{d}m$$

$$\geq \frac{\tilde{c}}{100\pi}\frac{\zeta^2\theta^2}{d}m - \frac{6\tilde{c}\zeta\theta^2}{\sqrt{d}}\sqrt{2m\ln\frac{m^2}{\delta}}$$

$$\geq \frac{\theta^2}{100\pi}\left[\frac{\tilde{c}\zeta^2 m}{d} - \sqrt{\frac{\tilde{c}\zeta^2 m}{d}}\right]$$

$$\geq \theta^2$$

where the first inequality is from the definition of event $E$, the second inequality is from that $\tilde{c} = \frac{1}{2(600\pi)^2\ln\frac{m^2}{\delta}}$, the third inequality is from that $\frac{\tilde{c}\zeta^2 m}{d} \geq 200\pi$.

Since $\cos\theta_0 \geq \cos\theta \geq 1 - \frac{1}{2}\theta^2$, this gives that $\cos\theta_m \geq 1 + \frac{1}{2}\theta^2 > 1$, contradiction.

Next, define $\tau := \min\left\{t \geq 0 : \theta_t \notin \left[\frac{1}{4}\theta, \frac{5}{3}\theta\right]\right\}$. We now know that $\tau \leq m$ by the reasoning above. It suffices to show that $\theta_\tau < \frac{1}{4}\theta$, that is, the first time when $\theta_t$ goes outside the interval $\left[\frac{1}{4}\theta, \frac{5}{3}\theta\right]$, it must be crossing the left boundary as opposed to the right one.

By the definition of $\tau$, for all $0 \leq t \leq \tau - 1$, $\theta_\tau \in \left[\frac{1}{4}\theta, \frac{5}{3}\theta\right]$. Thus,

$$\cos\theta_\tau - \cos\theta_0$$

$$= \sum_{t=0}^{\tau-1} D_t + \frac{\tilde{c}}{100\pi}\frac{\zeta^2\theta^2}{d}\tau$$

$$\geq \frac{\tilde{c}}{100\pi}\frac{\zeta^2\theta^2}{d}\tau - \frac{6\tilde{c}\zeta\theta^2}{\sqrt{d}}\sqrt{\tau\ln\frac{m^2}{\delta}}$$

$$\geq -900\pi\ln\frac{m^2}{\delta}\tilde{c}\theta^2 \geq -\frac{1}{75}\theta^2 \tag{3}$$

where the first inequality is by the definition of $E$; the second inequality is by minimizing over $\tau \in [0, m]$; the last inequality is from the definition of $\tilde{c}$.

Now, if $\theta_\tau \geq \frac{5}{3}\theta$, then

$$\cos\theta_\tau - \cos\theta_0 \leq \cos\frac{5}{3}\theta - \cos\theta$$

$$\leq 1 - \frac{1}{5}\left(\frac{5}{3}\right)^2\theta^2 - 1 + \frac{1}{2}\theta^2$$

$$< -\frac{1}{75}\theta^2$$

where the first inequality follows from $\theta_\tau \geq \frac{5}{3}\theta$ and $\theta_0 \leq \theta$, and the second inequality follows from Lemma 13. This contradicts with Inequality (3).

This gives that $\theta_\tau < \frac{5}{3}\theta$. Since $\theta_\tau \notin \left[\frac{1}{4}\theta, \frac{5}{3}\theta\right]$, it must be the case that $\theta_\tau < \frac{1}{4}\theta$. $\qquad\square$

2. We now show the following claim to conclude the proof.

**Claim 2.** $\theta_m$, *the angle in the last iteration, is at most* $\frac{1}{2}\theta$.

*Proof.* Define $\sigma = \max\left\{t \in [0, m] : \theta_t < \frac{1}{4}\theta\right\}$. by Claim 1, such $\sigma$ is well-defined on event $E$. We now show that $\theta_t$ will not exceed $\frac{1}{2}\theta$ afterwards. Assume for the sake of contradiction that for some $t > \sigma$, $\theta_t > \frac{1}{2}\theta$.

Now define $\gamma := \min\left\{t > \sigma : \theta_t > \frac{1}{2}\theta\right\}$. We know by the definitions of $\sigma$ and $\gamma$, for all $t \in [\sigma+1, \gamma-1]$, $\theta_t \in [\frac{1}{4}\theta, \frac{1}{2}\theta]$. Thus,

$$\cos\theta_\gamma - \cos\theta_{\sigma+1}$$

$$= \sum_{t=\sigma+1}^{\gamma-1} D_t + \frac{\tilde{c}}{100\pi}\frac{\zeta^2\theta^2}{d}(\gamma - \sigma - 1)$$

$$\geq \frac{\tilde{c}}{100\pi}\frac{\zeta^2\theta^2}{d}(\gamma - \sigma - 1) - \frac{6\tilde{c}\zeta\theta^2}{\sqrt{d}}\sqrt{(\gamma - \sigma - 1)\ln\frac{m^2}{\delta}}$$

$$\geq -900\pi\ln\frac{m^2}{\delta}\tilde{c} \geq -\frac{1}{75}\theta^2 \qquad (4)$$

where the first inequality is by the definition of $E$; the second inequality is by minimization over $\gamma - \sigma - 1 \in [0, m]$; the last inequality is from the definition of $\tilde{c}$.

On the other hand, $\theta_\gamma > \frac{1}{2}\theta$ and $\theta_\sigma < \frac{1}{4}\theta$. We have

$$\cos\theta_\gamma - \cos\theta_{\sigma+1} \leq \cos\theta_\gamma - \cos\theta_\sigma + \frac{6\tilde{c}\zeta\theta^2}{\sqrt{d}}$$

$$\leq \cos\frac{\theta}{2} - \cos\frac{\theta}{4} + \frac{6\tilde{c}\zeta\theta^2}{\sqrt{d}}$$

$$\leq 1 - \frac{1}{20}\theta^2 - 1 + \frac{1}{32}\theta^2 + \frac{6\tilde{c}\zeta\theta^2}{\sqrt{d}}$$

$$< -\frac{1}{75}\theta^2$$

where the first inequality follows from Lemma 6, the third follows from Lemma 13, and the last follows from algebra. This contradicts with Inequality (4). $\qquad\square$

Thus, with probability at least $1 - \delta/2$, $\theta_m \leq \frac{1}{2}\theta$. $\qquad\square$

## E  Progress Measure Analysis

In this section, we prove two key lemmas on $\cos\theta_t$ (Lemmas 9 and 10), our measure of progress. We show that under the bounded noise model and the adversarial noise model, $\cos\theta_t$ increases by a decent amount in expectation at each iteration of MODIFIED-PERCEPTRON, with appropriate settings of bandwidth $b$.

We begin with a generic lemma that gives a recurrence of $\cos\theta_t$ when the modified Perceptron update rule (1) is applied to a new example.

**Lemma 8.** *Suppose $w_t \in \mathbb{R}^d$ is a unit vector, and $(x_t, y_t)$ is an labeled example where $x_t \in \mathbb{R}^d$ is a unit vector and $y_t \in \{-1, +1\}$. Let $\theta_t = \theta(u, w_t)$. Then, update*

$$w_{t+1} \leftarrow w_t - 2\mathbb{1}\{y_t w_t \cdot x_t < 0\}(w_t \cdot x_t) \cdot x_t \qquad (5)$$

*gives an unit vector $w_{t+1}$ such that*

$$\cos\theta_{t+1} = \cos\theta_t - 2\mathbb{1}\{y_t w_t \cdot x_t < 0\}(w_t \cdot x_t) \cdot (u \cdot x_t) \qquad (6)$$

*Proof.* We first show that $w_{t+1}$ is still a unit vector. If $y_t = \text{sign}(w_t \cdot x_t)$, then $w_{t+1} = w_t$, thus it is still a unit vector; otherwise $w_{t+1} = w_t - 2(w_t \cdot x_t) \cdot x_t$. This gives that

$$\|w_{t+1}\|^2 = \|w_t\|^2 - 4(w_t \cdot x_t)(w_t \cdot x_t) + \|2(w_t \cdot x_t) \cdot x_t\|^2 = \|w_t\|^2 = 1.$$

This implies that $\cos\theta_t = w_t \cdot u$, and $\cos\theta_{t+1} = w_{t+1} \cdot u$. Now, taking inner products with $u$ on both sides of Equation (5), we get

$$w_{t+1} \cdot u = w_t \cdot u - 2\mathbb{1}\{y_t w_t \cdot x_t < 0\}(w_t \cdot x_t) \cdot (u \cdot x_t)$$

which is equivalent to Equation (6). $\qquad\square$

### E.1 Progress Measure under Bounded Noise

**Lemma 9** (Progress Measure under Bounded Noise). *Suppose $0 < \tilde{c} < \frac{1}{288}$, $b = \frac{\tilde{c}(1-2\eta)\theta}{\sqrt{d}}$, $\theta \leq \frac{27}{50}\pi$, and $(x_t, y_t)$ is drawn from $D|_{R_t}$, where $R_t = \left\{ (x, y) : x \cdot w_t \in [\frac{b}{2}, b] \right\}$. Meanwhile, the oracle $\mathcal{O}$ satisfies the $\eta$-bounded noise condition. If unit vector $w_t$ has angle $\theta_t$ with $u$ such that $\frac{1}{4}\theta \leq \theta_t \leq \frac{5}{3}\theta$, then update (5) has the following guarantee:*

$$\mathbb{E}\left[ \cos\theta_{t+1} - \cos\theta_t \mid \theta_t \right] \geq \frac{\tilde{c}}{100\pi} \frac{(1-2\eta)^2\theta^2}{d}.$$

*Proof.* Define random variable $\xi = x_t \cdot w_t$. By the tower property of conditional expectation, $\mathbb{E}\left[ \cos\theta_{t+1} - \cos\theta_t \mid \theta_t \right] = \mathbb{E}\left[ \mathbb{E}\left[ \cos\theta_{t+1} - \cos\theta_t \mid \theta_t, \xi \right] \mid \theta_t \right]$. Thus, it suffices to show

$$\mathbb{E}\left[ \cos\theta_{t+1} - \cos\theta_t \mid \theta_t, \xi \right] \geq \frac{\tilde{c}}{100\pi} \frac{(1-2\eta)^2\theta^2}{d}$$

for all $\theta_t \in [\frac{1}{4}\theta, \frac{5}{3}\theta]$ and $\xi \in [\frac{1}{2}b, b]$.

By Lemma 8, we know that

$$\cos\theta_{t+1} - \cos\theta_t = -2\mathbb{1}\left\{ y_t \neq \text{sign}(w_t \cdot x_t) \right\} (w_t \cdot x_t) \cdot (u \cdot x_t).$$

We simplify $\mathbb{E}\left[ \cos\theta_{t+1} - \cos\theta_t \mid \theta_t, \xi \right]$ as follows:

$$
\begin{aligned}
&\mathbb{E}\left[ \cos\theta_{t+1} - \cos\theta_t \mid \theta_t, \xi \right] \\
=\ & \mathbb{E}\left[ -2\xi u \cdot x_t \mathbb{1}\left\{ y_t = -1 \right\} \mid \theta_t, \xi \right] \\
=\ & \mathbb{E}\left[ -2\xi u \cdot x_t (\mathbb{1}\{u \cdot x_t > 0, y_t = -1\} + \mathbb{1}\{u \cdot x_t < 0, y_t = -1\}) \mid \theta_t, \xi \right] \\
\geq\ & \mathbb{E}\left[ -2\xi u \cdot x_t (\eta\mathbb{1}\{u \cdot x_t > 0\} + (1-\eta)\mathbb{1}\{u \cdot x_t < 0\}) \mid \theta_t, \xi \right] \\
=\ & \mathbb{E}\left[ -2\xi u \cdot x_t (\eta + (1-2\eta)\mathbb{1}\{u \cdot x_t < 0\}) \mid \theta_t, \xi \right] \\
=\ & -2\xi\left( \eta\mathbb{E}\left[ u \cdot x_t \mid \theta_t, \xi \right] + (1-2\eta)\mathbb{E}\left[ u \cdot x_t\mathbb{1}\{u \cdot x_t < 0\} \mid \theta_t, \xi \right] \right) &(7)
\end{aligned}
$$

where the second equality is from algebra, the first inequality is from that $\mathbb{P}[y_t = -1 | u \cdot x_t > 0] \leq \eta$ and $\mathbb{P}[y_t = -1 | u \cdot x_t < 0] \geq 1 - \eta$, the last two equalities are from algebra.

By Lemma 19 and that $0 \leq \theta_t \leq \frac{5}{3}\theta \leq \frac{9}{10}\pi$, $\mathbb{E}[u \cdot x_t | \theta_t, \xi] \leq \xi$ and $\mathbb{E}[u \cdot x_t\mathbb{1}\{u \cdot x_t < 0\} | \theta_t, \xi] \leq \xi - \frac{\theta_t}{36\sqrt{d}}$.

Thus,

$$
\begin{aligned}
&\mathbb{E}\left[ \cos\theta_{t+1} - \cos\theta_t \mid \theta_t, \xi \right] \\
\geq\ & -2\xi(\xi\eta + (\xi - \frac{\theta_t}{36\sqrt{d}})(1-2\eta)) \\
\geq\ & 2\xi(\frac{\theta_t}{36\sqrt{d}}(1-2\eta) - \xi) \\
\geq\ & b\frac{\theta_t}{72\sqrt{d}}(1-2\eta) \\
\geq\ & \frac{\tilde{c}}{100\pi} \frac{(1-2\eta)^2\theta^2}{d}
\end{aligned}
$$

where the first and second inequalities are from algebra, the third inequality is from that $\xi \leq b \leq \frac{\theta(1-2\eta)}{288\sqrt{d}} \leq \frac{\theta_t(1-2\eta)}{72\sqrt{d}}$, and that $\xi \geq \frac{b}{2}$. the last inequality is by expanding $b = \frac{\tilde{c}(1-2\eta)\theta}{\sqrt{d}}$ and that $\theta_t \geq \frac{\theta}{4}$.

In conclusion, if $\frac{1}{4}\theta \leq \theta_t \leq \frac{5}{3}\theta$, then $\mathbb{E}\left[ \cos\theta_{t+1} - \cos\theta_t \mid \theta_t, \xi \right] \geq \frac{\tilde{c}}{100\pi} \frac{(1-2\eta)^2\theta^2}{d}$ for $\xi \in [\frac{b}{2}, b]$. The lemma follows. $\square$

## E.2 Progress Measure under Adversarial Noise

**Lemma 10** (Progress Measure under Adversarial Noise). *Suppose $0 \leq \tilde{c} \leq \frac{1}{100\pi}$, $b = \frac{\tilde{c}\theta}{\sqrt{d}}$, $\theta \leq \frac{27}{50}\pi$, and $(x_t, y_t)$ is drawn from distribution $D|_{R_t}$ where $R_t = \left\{(x, y) : x \cdot w_t \in [\frac{b}{2}, b]\right\}$. Meanwhile, the oracle $\mathcal{O}$ satisfies the $\nu$-adversarial noise condition where $\nu \leq \frac{\tilde{c}\theta}{192(200\pi)^2}$. If unit vector $w_t$ has angle $\theta_t$ with $u$ such that $\frac{1}{4}\theta \leq \theta_t \leq \frac{5}{3}\theta$, then update (5) has the following guarantee:*

$$\mathbb{E}\left[\cos\theta_{t+1} - \cos\theta_t \mid \theta_t\right] \geq \frac{\tilde{c}}{100\pi}\frac{\theta^2}{d}.$$

*Proof.* Define random variable $\xi = x_t \cdot w_t$.

By Lemma 8, we know that

$$\cos\theta_{t+1} - \cos\theta_t = -2\mathbb{1}\left\{y_t \neq \text{sign}(w_t \cdot x_t)\right\}(w_t \cdot x_t) \cdot (u \cdot x_t).$$

We expand $\mathbb{E}\left[\cos\theta_{t+1} - \cos\theta_t \mid \theta_t\right]$ as follows.

$$
\begin{aligned}
&\mathbb{E}\left[\cos\theta_{t+1} - \cos\theta_t \mid \theta_t\right] \\
=\ &\mathbb{E}\left[-2(w_t \cdot x_t)(u \cdot x_t)\mathbb{1}\left\{y_t = -1\right\} \mid \theta_t\right] \\
=\ &\mathbb{E}\left[-2(w_t \cdot x_t)(u \cdot x_t)\mathbb{1}\left\{u \cdot x_t < 0\right\} \mid \theta_t\right] \\
&+\mathbb{E}\left[2(w_t \cdot x_t)(u \cdot x_t)(\mathbb{1}\left\{y_t = +1, u \cdot x_t < 0\right\} - \mathbb{1}\left\{y_t = -1, u \cdot x_t > 0\right\})) \mid \theta_t\right] \quad (8)
\end{aligned}
$$

We bound the two terms separately. Firstly,

$$
\begin{aligned}
&\mathbb{E}\left[-2(w_t \cdot x_t)(u \cdot x_t)\mathbb{1}\left\{u \cdot x_t < 0\right\} \mid \theta_t\right] \\
\geq\ &-b\mathbb{E}\left[(u \cdot x_t)\mathbb{1}\left\{u \cdot x_t < 0\right\} \mid \theta_t\right] \\
=\ &-b\mathbb{E}\left[\mathbb{E}\left[(u \cdot x_t)\mathbb{1}\left\{u \cdot x_t < 0\right\} \mid \theta_t, b\right] \mid \theta_t\right] \\
\geq\ &b(\frac{\theta_t}{36\sqrt{d}} - b) \quad (9)
\end{aligned}
$$

where the first inequality is from that $-(u \cdot x_t)\mathbb{1}\left\{u \cdot x_t < 0\right\} \geq 0$ and $w_t \cdot x_t \geq \frac{b}{2}$, the equality is from the tower property of conditional expectation, the second inequality is from Lemma 19.

Secondly,

$$
\begin{aligned}
&\left|\mathbb{E}\left[2(w_t \cdot x_t)(u \cdot x_t)(\mathbb{1}\left\{y_t = +1, u \cdot x_t < 0\right\} - \mathbb{1}\left\{y_t = -1, u \cdot x_t > 0\right\})) \mid \theta_t\right]\right| \\
\leq\ &2b\mathbb{E}\left[|u \cdot x_t|\mathbb{1}\left\{y_t \neq \text{sign}(u \cdot x_t))\right\} \mid \theta_t\right] \\
\leq\ &2b\sqrt{\mathbb{E}\left[\mathbb{1}\left\{y_t \neq \text{sign}(u \cdot x_t))\right\} \mid \theta_t\right] \cdot \mathbb{E}\left[(u \cdot x_t)^2 \mid \theta_t\right]} \\
=\ &2b\sqrt{\mathbb{P}\left[y_t \neq \text{sign}(u \cdot x_t) \mid \theta_t\right]\mathbb{E}\left[\mathbb{E}\left[(u \cdot x_t)^2 \mid \theta_t, \xi\right] | \theta_t\right]} \quad (10)
\end{aligned}
$$

where the first inequality is from that $|\mathbb{E}[X]| \leq \mathbb{E}|X|$, and $w_t \cdot x_t \leq b$, the second inequality is from Cauchy-Schwarz, the third equality is by algebra.

Now we look at the two terms inside the square root. First,

$$
\begin{aligned}
&\mathbb{P}\left[y_t \neq \text{sign}(u \cdot x_t) \mid \theta_t\right] \\
=\ &\mathbb{P}_{x \sim D|_{R_t}}\left[y \neq \text{sign}(u \cdot x)\right] \\
\leq\ &\frac{\mathbb{P}_{(x,y) \sim D}\left[y \neq \text{sign}(u \cdot x)\right]}{\mathbb{P}_{x \sim D}\left[x_1 \in [b/2, b]\right]} \\
\leq\ &\frac{8\pi\nu}{\tilde{c}\theta} \\
\leq\ &\frac{1}{16(200\pi)^2}
\end{aligned}
$$

where the first inequality is from that $\mathbb{P}[A|B] \leq \frac{\mathbb{P}[A]}{\mathbb{P}[B]}$, the second inequality is from Lemma 18 that $\mathbb{P}_{x \sim D}\left[x_1 \in [b/2, b]\right] \geq \frac{\sqrt{d}}{8\pi}b = \frac{\tilde{c}\theta}{8\pi}$, and the last inequality is by our assumption on $\nu$.

Second, fix $\xi \in [\frac{b}{2}, b]$, $\xi \leq b \leq \frac{\theta_t}{4\sqrt{d}}$. Item 2 of Lemma 19 implies that $\mathbb{E}\left[(u \cdot x_t)^2 \mid \theta_t, \xi\right] \leq \frac{5\theta_t^2}{d}$. By the tower property of conditional expectation, $\mathbb{E}\left[(u \cdot x_t)^2 \mid \theta_t\right] \leq \frac{5\theta_t^2}{d}$. Continuing Equation (10), we get

$$\left|\mathbb{E}\left[2(w_t \cdot x_t)(u \cdot x_t)(\mathbb{1}\left\{y_t = +1, u \cdot x_t < 0\right\} - \mathbb{1}\left\{y_t = -1, u \cdot x_t > 0\right\}\right) \mid \theta_t\right]\right| \leq b\frac{\theta_t}{100\pi\sqrt{d}}.$$

(11)

Continuing Equation (8), we have

$$\mathbb{E}\left[\cos\theta_{t+1} - \cos\theta_t \mid \theta_t\right]$$
$$\geq \quad b(\frac{\theta_t}{36\sqrt{d}} - \frac{\theta_t}{100\pi\sqrt{d}} - b)$$
$$\geq \quad b\frac{\theta_t}{25\pi\sqrt{d}} \geq \frac{\tilde{c}}{100\pi}\frac{\theta^2}{d}$$

where the first inequality is from Equations (9) and (11), the second inequality is from algebra and that $b \leq \frac{\theta_t}{100\pi\sqrt{d}}$, the third inequality is by expanding $b = \frac{\tilde{c}\theta}{\sqrt{d}}$ and $\theta_t \geq \frac{\theta}{4}$. □

# F   Acute Initialization

We show in this section that the angle between the initial vector $v_0$ and the underlying halfspace $u$ can be assumed to be acute under the two noise settings without loss of generality. To this end, we give two algorithms (Algorithms 5 and 6) that returns a halfspace that has angle at most $\frac{\pi}{4}$ with $u$, with constant overhead in label and time complexities. The techniques here are due to Appendix B of [6]. This fact, in conjunction with Theorems 2 and 3, yield an active learning algorithm that learns the target halfspace unconditionally with a constant overhead of label and time complexities.

For the bounded noise setting, we construct Algorithm 5 as an initialization procedure. It runs ACTIVE-PERCEPTRON twice, taking a vector $v_0$ and its opposite direction $-v_0$ as initializers. Then it performs hypothesis testing using $\tilde{O}(\frac{1}{(1-2\eta)^2})$ labeled examples to identify a halfspace that has angle at most $\frac{\pi}{4}$ with $u$.

---

**Algorithm 5** Master Algorithm in the Bounded Noise Setting

---

**Input:** Labeling oracle $\mathcal{O}$, confidence $\delta$, noise upper bound $\eta$, sample schedule $\{m_k\}$, band width $\{b_k\}$.
**Output:** a halfspace $\hat{v}$ such that $\theta(\hat{v}, u) \leq \frac{\pi}{4}$.
  1: $v_0 \leftarrow (1, 0, \ldots, 0)$.
  2: $v_+ \leftarrow$ ACTIVE-PERCEPTRON$(\mathcal{O}, v_0, \frac{(1-2\eta)}{16}, \frac{\delta}{3}, \{m_k\}, \{b_k\})$.
  3: $v_- \leftarrow$ ACTIVE-PERCEPTRON$(\mathcal{O}, -v_0, \frac{(1-2\eta)}{16}, \frac{\delta}{3}, \{m_k\}, \{b_k\})$.
  4: Define region $R := \left\{x : \text{sign}(v_+ \cdot x) \neq \text{sign}(v_- \cdot x)\right\}$.
  5: $S \leftarrow$ Draw $\frac{8}{(1-2\eta)^2} \ln\frac{6}{\delta}$ iid examples from $D|_R$ and query their labels.
  6: **if** $\text{err}_S(h_{v_+}) \leq \text{err}_S(h_{v_-})$ **then**
  7:     **return** $v_+$
  8: **else**
  9:     **return** $v_-$
10: **end if**

---

**Theorem 6.** *Suppose Algorithm 5 has inputs labeling oracle $\mathcal{O}$ that satisfies $\eta$-bounded noise condition with respect to $u$, confidence $\delta$, sample schedule $\{m_k\}$ where $m_k = \Theta\left(\frac{d}{(1-2\eta)^2}\left(\ln\frac{d}{(1-2\eta)^2} + \ln\frac{k}{\delta}\right)\right)$, band width $\{b_k\}$ where $b_k = \tilde{\Theta}\left(\frac{2^{-k}(1-2\eta)}{\sqrt{d}}\right)$. Then, with probability at least $1 - \delta$, the output $\hat{v}$ is such that $\theta(\hat{v}, u) \leq \frac{\pi}{4}$. Furthermore, (1) the total number of label*

*queries to oracle $\mathcal{O}$ is at most $\tilde{O}\left(\frac{d}{(1-2\eta)^2}\right)$; (2) the total number of unlabeled examples drawn is $\tilde{O}\left(\frac{d}{(1-2\eta)^3}\right)$; (3) the algorithm runs in time $\tilde{O}\left(\frac{d^2}{(1-2\eta)^3}\right)$.*

*Proof.* Note that one of $\theta(v_0, u)$, $\theta(-v_0, u)$ is at most $\frac{\pi}{2}$. From Theorem 2 and union bound, we know that with probability at least $1 - \frac{2\delta}{3}$, either $\theta(v_+, u) \leq \frac{(1-2\eta)\pi}{16}$, or $\theta(v_-, u) \leq \frac{(1-2\eta)\pi}{16}$.

Suppose without loss of generality, $\theta(v_+, u) \leq \frac{(1-2\eta)\pi}{16}$. We consider two cases.

**Case 1:** $\theta(v_+, v_-) \leq \pi/8$. By triangle inequality, $\theta(v_-, u) \leq \theta(v_+, u) + \theta(v_+, v_-) \leq \pi/4$. In this case, $\theta(v_+, u) \leq \frac{\pi}{4}$ and $\theta(v_-, u) \leq \frac{\pi}{4}$ holds simultaneously. Therefore, the returned vector $\hat{v}$ satisfies $\theta(\hat{v}, u) \leq \frac{\pi}{4}$.

**Case 2:** $\theta(v_+, v_-) > \pi/8$. In this case, $\mathbb{P}[x \in R] \geq 1/8$, thus,

$$\mathbb{P}_R[\text{sign}(v_+ \cdot x) \neq \text{sign}(u \cdot x)] \leq \frac{\mathbb{P}[\text{sign}(v_+ \cdot x) \neq \text{sign}(u \cdot x)]}{\mathbb{P}[x \in R]} \leq \frac{1 - 2\eta}{8} = \frac{1}{4}(\frac{1}{2} - \eta).$$

Meanwhile, $\mathbb{P}_R[\text{sign}(v_+ \cdot x) \neq y] \leq \eta \mathbb{P}_R[\text{sign}(v_+ \cdot x) = \text{sign}(u \cdot x)] + \mathbb{P}_R[\text{sign}(v_+ \cdot x) \neq \text{sign}(u \cdot x)]$. Therefore,

$$
\begin{aligned}
&\frac{1}{2} - \mathbb{P}_R[\text{sign}(v_+ \cdot x) \neq y] \\
\geq{}& (\frac{1}{2} - \eta)\mathbb{P}_R[\text{sign}(v_+ \cdot x) = \text{sign}(u \cdot x)] - \frac{1}{2}\mathbb{P}_R[\text{sign}(v_+ \cdot x) \neq \text{sign}(u \cdot x)] \\
\geq{}& (\frac{1}{2} - \eta) \cdot \frac{1}{2} - (\frac{1}{2} - \eta) \cdot \frac{1}{4} \\
\geq{}& \frac{1}{4}(\frac{1}{2} - \eta)
\end{aligned}
$$

Since $v_+$ disagrees with $v_-$ everywhere on $R$, $\mathbb{P}_R[\text{sign}(v_+ \cdot x) \neq y] + \mathbb{P}_R[\text{sign}(v_- \cdot x) \neq y] = 1$. Thus, $\text{err}_{D|R}(h_{v_+}) \leq \frac{1}{2} - (\frac{1}{2} - \eta)\frac{1}{4}$ and $\text{err}_{D|R}(h_{v_-}) \geq \frac{1}{2} + (\frac{1}{2} - \eta)\frac{1}{4}$. Therefore, by Hoeffding's Inequality, with probability at least $1 - \delta/3$,

$$\text{err}_S(v_+) < \frac{1}{2} < \text{err}_S(v_-)$$

therefore $v_+$ will be selected for $\hat{v}$. This shows that $\theta(\hat{v}, u) \leq \pi/4$.

In conclusion, by union bound, we have shown that with probability $1 - \delta$, $\theta(\hat{v}, u) \leq \frac{\pi}{4}$. The label complexity, unlabeled sample complexity, and time complexity of the algorithm follows immediately from Theorem 2. □

For the adversarial noise setting, [6] outlines an algorithm that returns a vector that has angle at most $\frac{\pi}{4}$ with $u$. We state the algorithm in our context for completeness.

**Theorem 7.** *Suppose Algorithm 6 has inputs labeling oracle $\mathcal{O}$ that satisfies $\eta$-bounded noise condition with respect to $u$, confidence $\delta$, sample schedule $\{m_k\}$ where $m_k = \Theta\left(d(\ln d + \ln \frac{k}{\delta})\right)$, band width $\{b_k\}$ where $b_k = \tilde{\Theta}\left(\frac{2^{-k}}{\sqrt{d}}\right)$. Then, with probability at least $1 - \delta$, the output $\hat{v}$ is such that $\theta(\hat{v}, u) \leq \frac{\pi}{4}$. Furthermore, (1) the total number of label queries to oracle $\mathcal{O}$ is at most $\tilde{O}(d)$; (2) the total number of unlabeled examples drawn is $\tilde{O}(d)$; (3) the algorithm runs in time $\tilde{O}(d^2)$.*

The proof of this theorem is almost the same as Theorem 6 and is thus omitted.

# G Basic Lemmas for the Upper Bounds

In this section, we present a few useful lemmas that serve as the basis of proving Theorems 2 and 3.

---

**Algorithm 6** Master Algorithm in the Adversarial Noise Setting

---

**Input:** Labeling oracle $\mathcal{O}$, confidence $\delta$

**Output:** a halfspace $\hat{v}$ such that $\theta(\hat{v}, u) \leq \frac{\pi}{4}$.

1: $v_0 \leftarrow (1, 0, \ldots, 0)$.
2: $v_+ \leftarrow \text{ACTIVE-PERCEPTRON}(\mathcal{O}, v_0, \frac{1}{16}, \frac{\delta}{3}, \{m_k\}, \{b_k\})$.
3: $v_- \leftarrow \text{ACTIVE-PERCEPTRON}(\mathcal{O}, -v_0, \frac{1}{16}, \frac{\delta}{3}, \{m_k\}, \{b_k\})$.
4: Define region $R := \{x : \text{sign}(v_+ \cdot x) \neq \text{sign}(v_- \cdot x)\}$.
5: $S \leftarrow$ Draw $8 \ln \frac{6}{\delta}$ iid examples from $D|_R$ and query their labels.
6: **if** $\text{err}_S(h_{v_+}) \leq \text{err}_S(h_{v_-})$ **then**
7:     **return** $v_+$
8: **else**
9:     **return** $v_-$
10: **end if**

---

### G.1 Basic Facts

We first collect a few useful facts for algebraic manipulations.

**Lemma 11.** *If* $0 \leq x \leq 1 - \frac{1}{e}$, *then for any* $d \geq 1$, $(1 - \frac{x}{d})^{\frac{d}{2}} \geq e^{-x} \geq \frac{1}{2}$.

**Lemma 12.** *Given* $a \in (0, \pi)$, *if* $x \in [0, a]$, *then* $\frac{\sin a}{a} x \leq \sin x \leq x$.

**Lemma 13.** *If* $x \in [0, \pi]$, *then* $1 - \frac{x^2}{2} \leq \cos x \leq 1 - \frac{x^2}{5}$.

**Lemma 14.** *Let* $\text{B}(x, y) = \int_0^1 (1 - t)^{x-1} t^{y-1} dt$ *be the Beta function. Then* $\frac{2}{\sqrt{d-1}} \leq \text{B}(\frac{1}{2}, \frac{d}{2}) \leq \frac{\pi}{\sqrt{d}}$.

### G.2 Probability Inequalities

**Lemma 15** (Azuma's Inequality). *Let* $\{Y_t\}_{t=1}^m$ *be a bounded submartingale difference sequence, that is,* $\mathbb{E}[Y_t | Y_1, \ldots, Y_{t-1}] \geq 0$, *and* $|Y_t| \leq \sigma$. *Then, with probability at least* $1 - \delta$,

$$\sum_{t=1}^m Y_t \geq -\sigma \sqrt{2m \ln \frac{1}{\delta}}$$

**Lemma 16** (Concentration of Geometric Random Variables). *Suppose* $Z_1, \ldots, Z_n$ *are iid geometric random variables with parameter* $p$. *Then,*

$$\mathbb{P}[Z_1 + \ldots + Z_n > \frac{2n}{p}] \leq \exp(-\frac{n}{4})$$

*Proof.* Since $Z_1 + \ldots + Z_n > \frac{2n}{p}$ implies that $Z_1 + \ldots + Z_n \geq \lceil \frac{2n}{p} \rceil$ (as $Z_1 + \ldots + Z_n$ is an integer), the left hand side is at most $\mathbb{P}[Z_1 + \ldots + Z_n \geq \lceil \frac{2n}{p} \rceil]$.

Let $X_1, \ldots, X_{\lceil \frac{2n}{p} \rceil}$ be a sequence of iid $\text{Bernoulli}(p)$ random variables. By standard relationship between Bernoulli random variables and geometric random variables, we have that

$$\mathbb{P}[Z_1 + \ldots + Z_n \geq \lceil \frac{2n}{p} \rceil] = \mathbb{P}[X_1 + \ldots + X_{\lceil \frac{2n}{p} \rceil - 1} \leq n - 1]$$

Note that $\mathbb{P}[X_1 + \ldots + X_{\lceil \frac{2n}{p} \rceil - 1} \leq n - 1] \leq \mathbb{P}[X_1 + \ldots + X_{\lceil \frac{2n}{p} \rceil} \leq n]$ since $X_{\lceil \frac{2n}{p} \rceil} \leq 1$. Applying Chernoff bound, the above probability is at most $\exp(-\lceil \frac{2n}{p} \rceil \cdot p \cdot \frac{1}{8}) \leq \exp(-\frac{n}{4})$. $\square$

### G.3 Properties of the Uniform Distribution over the Unit Sphere

**Lemma 17** (Marginal Density and Conditional Density). *If* $(x_1, x_2, \ldots, x_d)$ *is drawn from the uniform distribution over the unit sphere, then:*

1. $(x_1, x_2)$ *has a density function of* $p(z_1, z_2)$, *where* $p(z_1, z_2) = \frac{(1 - z_1^2 - z_2^2)^{\frac{d-4}{2}}}{\frac{2\pi}{d-2}}$.

2. *Conditioned on* $x_2 = b$, $x_1$ *has a density function of* $p_b(z)$, *where* $p_b(z) = \frac{(1-b^2-z^2)^{\frac{d-4}{2}}}{(1-b^2)^{\frac{d-3}{2}} \, \mathrm{B}(\frac{d-2}{2},\frac{1}{2})}$.

3. $x_1$ *has a density function of* $p(z)$, *where* $p(z) = \frac{(1-z^2)^{\frac{d-3}{2}}}{\mathrm{B}(\frac{d-1}{2},\frac{1}{2})}$.

**Lemma 18.** *Suppose $x$ is drawn uniformly from the unit sphere, and $b \leq \frac{1}{10\sqrt{d}}$. Then,*

$$\mathbb{P}\left[x_1 \in \left[\frac{b}{2},b\right]\right] \geq \frac{\sqrt{d}}{8\pi}b.$$

*Proof.*

$$\mathbb{P}\left[x_1 \in \left[\frac{b}{2},b\right]\right]$$

$$= \frac{\int_{b/2}^{b}(1-t^2)^{\frac{d-3}{2}}\,\mathrm{d}t}{\mathrm{B}(\frac{d-1}{2},\frac{1}{2})}$$

$$\geq \frac{\frac{b}{2}(1-b^2)^{\frac{d-3}{2}}}{\frac{\pi}{\sqrt{d-1}}} \geq \frac{\sqrt{d}}{8\pi}b$$

where the first equality is from item 3 of Lemma 17, giving the exact probability density function of $x_1$, the first inequality is from that $(1-t^2)^{\frac{d-3}{2}} \geq (1-b^2)^{\frac{d-3}{2}}$ when $t \in \left[b/2,b\right]$, and Lemma 14 giving upper bound on $\mathrm{B}(\frac{d-1}{2},\frac{1}{2})$, and the second inequality is from Lemma 11 and that $d-1 \geq \frac{d}{2}$. $\quad\square$

**Lemma 19.** *Suppose $x$ is drawn uniformly from unit sphere restricted to the region $\{x : v \cdot x = \xi\}$, and $u,v$ are unit vectors such that $\theta(u,v) = \theta \in [0,\frac{9}{10}\pi]$ and $0 \leq \xi \leq \frac{\theta}{4\sqrt{d}}$. Then,*

1. $\mathbb{E}[u \cdot x] \leq \xi$.

2. $\mathbb{E}[(u \cdot x)^2] \leq \frac{5\theta^2}{d}$.

3. $\mathbb{E}[(u \cdot x)\mathbb{1}\{u \cdot x < 0\}] \leq \xi - \frac{\theta}{36\sqrt{d}}$.

*Proof.* By spherical symmetry, without loss of generality, let $v = (0,1,0,\ldots,0)$, and $u = (\sin\theta,\cos\theta,0,\ldots,0)$. Let $x = (x_1,\ldots,x_d)$.

1.

$$\mathbb{E}[u \cdot x]$$
$$= \mathbb{E}[x_1\sin\theta + x_2\cos\theta|x_2 = \xi]$$
$$= \mathbb{E}[x_1|x_2 = \xi]\sin\theta + \xi\cos\theta$$
$$\leq \xi$$

where the first two equalities are by algebra, the inequality follows from $\cos\theta \leq 1$ and $\mathbb{E}[x_1|x_2 = \xi] = 0$ since the conditional distribution of $x_1$ given $x_2 = \xi$ is symmetric around the origin.

2.

$$
\begin{aligned}
&\mathbb{E}[(u \cdot x)^2] \\
={}& \mathbb{E}[(x_1 \sin\theta + x_2 \cos\theta)^2 | x_2 = \xi] \\
\leq{}& \mathbb{E}[2x_1^2 \sin^2\theta + 2x_2^2 \cos^2\theta | x_2 = \xi] \\
\leq{}& 2\mathbb{E}[x_1^2 | x_2 = \xi]\sin^2\theta + 2\xi^2 \\
\leq{}& 2\theta^2 \frac{\int_{-1}^1 z^2 (1-z^2)^{\frac{d-4}{2}}\,\mathrm{d}z}{\mathrm{B}(\frac{d-2}{2}, \frac{1}{2})} + 2\xi^2 \\
={}& 2\theta^2 \frac{\mathrm{B}(\frac{d-2}{2}, \frac{3}{2})}{\mathrm{B}(\frac{d-2}{2}, \frac{1}{2})} + 2\xi^2 \\
\leq{}& \frac{5\theta^2}{d}
\end{aligned}
$$

where the first equality is by definition of $u$, the first inequality is from algebra that $(A + B)^2 \leq 2A^2 + 2B^2$, the second inequality is from that $|\cos\theta| \leq 1$, the third inequality is from item 2 of Lemma 17 and that $\sin\theta \leq \theta$, and the last inequality is from the fact that $\frac{\mathrm{B}(\frac{d-2}{2}, \frac{3}{2})}{\mathrm{B}(\frac{d-2}{2}, \frac{1}{2})} = \frac{1}{d-1} \leq \frac{2}{d}$, and $\xi^2 \leq \frac{\theta^2}{16d}$.

3.

$$
\begin{aligned}
&\mathbb{E}[(u \cdot x)\mathbb{1}\{u \cdot x < 0\}] \\
={}& \mathbb{E}[(x_1 \sin\theta + x_2 \cos\theta)\mathbb{1}\{x_1 < -\xi\cot\theta\} | x_2 = \xi] \\
\leq{}& \mathbb{E}[x_1 \mathbb{1}\{x_1 < -\xi\cot\theta\} | x_2 = \xi]\sin\theta + \xi \\
={}& \xi + \sin\theta \int_{-\sqrt{1-\xi^2}}^{-\xi\cot\theta} \frac{(1-\xi^2-x_1^2)^{\frac{d-4}{2}} x_1}{(1-\xi^2)^{\frac{d-3}{2}} \mathrm{B}(\frac{d-2}{2}, \frac{1}{2})}\,\mathrm{d}x_1 \\
={}& \xi - \sin\theta \frac{\frac{2}{d-2}\left(1-\left(\frac{\xi}{\sin\theta}\right)^2\right)^{\frac{d-2}{2}}}{(1-\xi^2)^{\frac{d-3}{2}} \mathrm{B}(\frac{d-2}{2}, \frac{1}{2})} \\
\leq{}& \xi - \sin\theta \frac{2}{\pi\sqrt{d-2}}\left(1-\left(\frac{\xi}{\sin\theta}\right)^2\right)^{\frac{d-2}{2}} \\
\leq{}& \xi - \frac{\sin\theta}{\pi\sqrt{d}} \\
\leq{}& \xi - \frac{\theta}{36\sqrt{d}}
\end{aligned}
$$

where the first inequality is by algebra and $|\cos\theta| \leq 1$, the second equality is by item 2 of Lemma 17, the third equality is by integration, the second inequality is from $(1-\xi^2)^{\frac{d-3}{2}} \leq 1$ and Lemma 14 that $\mathrm{B}(\frac{d-2}{2}, \frac{1}{2}) \leq \frac{\pi}{\sqrt{d-2}}$, the third inequality follows by Lemma 11 that $\left(1-\left(\frac{\xi}{\sin\theta}\right)^2\right)^{\frac{d-2}{2}} \geq \frac{1}{2}$, since $\xi \leq \frac{\theta}{4\sqrt{d}}$, and the last inequality follows from Lemma 12 that $\sin\theta \geq \frac{5\theta}{18\pi}$ when $\theta \in [0, \frac{9}{10}\pi]$ and algebra.

$\square$

## H  Proof of the Lower Bound

In this section, we give the proof of Theorem 1 (label complexity lower bound in the bounded noise setting). The proof follows from two key lemmas, Lemma 24 and Lemma 25. We start with some additional definitions.

**Definition 3.** *Let $\mathbb{P}, \mathbb{Q}$ be two probability measures on a common measurable space and $\mathbb{P}$ is absolutely continuous with respect to $\mathbb{Q}$.*

- *The KL-divergence between $\mathbb{P}$ and $\mathbb{Q}$ is defined as $D_{KL}(\mathbb{P}, \mathbb{Q}) = \mathbb{E}_{X \sim \mathbb{P}} \ln \frac{\mathbb{P}(X)}{\mathbb{Q}(X)}$.*

- *We define $d_{KL}(p, q) = D_{KL}(\mathbb{P}, \mathbb{Q})$, where $\mathbb{P}, \mathbb{Q}$ are distributions of a Bernoulli(p) and a Bernoulli(q) random variables respectively.*

- *For random variables $X, Y, Z$, define the mutual information between $X$ and $Y$ under $\mathbb{P}$ as $I(X; Y) = D_{KL}(\mathbb{P}(X, Y), \mathbb{P}(X)\mathbb{P}(Y)) = \mathbb{E}_{X,Y} \ln \frac{\mathbb{P}(X,Y)}{\mathbb{P}(X)P(Y)}$, and define the mutual information between $X$ and $Y$ conditioned on $Z$ under $\mathbb{P}$ as $I(X; Y \mid Z) = \mathbb{E}_{X,Y,Z} \ln \frac{\mathbb{P}(X,Y|Z)}{\mathbb{P}(X|Z)P(Y|Z)}$.*

- *For a sequence of random variables $X_1, X_2, \ldots$, denote by $X^n$ the subsequence $\{X_1, X_2, \ldots X_n\}$.*

We will use the following two folklore information-theoretic lower bounds.

**Lemma 20.** *Let $\mathcal{W}$ be a class of parameters, and $\{P_w : w \in \mathcal{W}\}$ be a class of probability distributions indexed by $\mathcal{W}$ over some sample space $\mathcal{X}$. Let $d : \mathcal{W} \times \mathcal{W} \to \mathbb{R}$ be a semi-metric. Let $\mathcal{V} = \{w_1, \ldots, w_M\} \subseteq \mathcal{W}$ such that $\forall i \neq j$, $d(w_i, w_j) \geq 2s > 0$. Let $V$ be a random variable uniformly taking values from $\mathcal{V}$, and $X$ be drawn from $P_V$. Then for any algorithm $\mathcal{A}$ that given a sample $X$ drawn from $P_w$ outputs $\mathcal{A}(X) \in \mathcal{W}$, the following inequality holds:*

$$\sup_{w \in \mathcal{W}} P_w \big( d(w, \mathcal{A}(X)) \geq s \big) \geq 1 - \frac{I(V; X) + \ln 2}{\ln M}$$

*Proof.* For any algorithm $\mathcal{A}$, define a test function $\hat{\Psi} : \mathcal{X} \to \{1, \ldots, M\}$ such that

$$\hat{\Psi}(X) = \arg \min_{i \in \{1, \ldots, M\}} d(\mathcal{A}(X), w_i)$$

We have

$$\sup_{w \in \mathcal{W}} P_w \big( d(w, \mathcal{A}(X)) \geq s \big) \geq \max_{w \in \mathcal{V}} P_w \big( d(w, \mathcal{A}(X)) \geq s \big) \geq \max_{i \in \{1, \ldots, M\}} P_{w_i} \big( \hat{\Psi}(X) \neq i \big)$$

The desired result follows by classical Fano's Inequality:

$$\max_{i \in \{1, \ldots, M\}} P_{w_i} \big( \hat{\Psi}(X) \neq i \big) \geq 1 - \frac{I(V; X) + \ln 2}{\ln M}$$

$\square$

**Lemma 21.** *[4, Lemma 5.1] Let $\gamma \in (0, 1)$, $\delta \in (0, \frac{1}{4})$, $p_0 = \frac{1-\gamma}{2}$, $p_1 = \frac{1+\gamma}{2}$. Suppose that $\alpha \sim$ Bernoulli$(\frac{1}{2})$ is a random variable, $\xi_1, \ldots, \xi_m$ are i.i.d. (given $\alpha$) Bernoulli$(p_\alpha)$ random variables. If $m \leq 2 \left\lfloor \frac{1-\gamma^2}{2\gamma^2} \ln \frac{1}{8\delta(1-2\delta)} \right\rfloor$, then for any function $f : \{0, 1\}^m \to \{0, 1\}$, $\mathbb{P}\big( f(\xi_1, \ldots, \xi_m) \neq \alpha \big) > \delta$.*

Next, we present two technical lemmas.

**Lemma 22.** *[48, Lemma 6] For any $0 < \gamma \leq \frac{1}{2}$, $d \geq 1$, there is a finite set $\mathcal{V} \in \mathbb{S}^{d-1}$ such that the following two statements hold:*

*1. For any distinct $w_1, w_2 \in \mathcal{V}$, $\theta(w_1, w_2) \geq \pi\gamma$;*

*2. $|\mathcal{V}| \geq \frac{\sqrt{d}}{2} \left( \frac{1}{2\pi\gamma} \right)^{d-1} - 1$.*

**Lemma 23.** *If $p \in [0, 1]$ and $q \in (0, 1)$, then $d_{KL}(p, q) \leq \frac{(p-q)^2}{q(1-q)}$.*

*Proof.*

$$\begin{aligned}
d_{\text{KL}}(p, q) &=& p \ln \frac{p}{q} + (1-p) \ln \frac{1-p}{1-q} \\
&\le& p(\frac{p}{q} - 1) + (1-p)(\frac{1-p}{1-q} - 1) \\
&=& \frac{(p-q)^2}{q(1-q)}
\end{aligned}$$

where the inequality follows by $\ln x \le x - 1$. $\qquad\square$

**Lemma 24.** *For any $0 \le \eta < \frac{1}{2}$, $d > 4$, $0 < \epsilon \le \frac{1}{4\pi}$, $0 < \delta < \frac{1}{2}$, for any active learning algorithm $\mathcal{A}$, there is a $u \in \mathbb{S}^{d-1}$, and a labeling oracle $\mathcal{O}$ that satisfies $\eta$-bounded noise condition with respect to $u$, such that if with probability at least $1 - \delta$, $\mathcal{A}$ makes at most $n$ queries to $\mathcal{O}$ and outputs $v \in \mathbb{S}^{d-1}$ such that $\mathbb{P}[\text{sign}(v \cdot x) \ne \text{sign}(u \cdot x)] \le \epsilon$, then $n \ge \frac{d \ln \frac{1}{\epsilon}}{16(1-2\eta)^2}$.*

*Proof.* We will prove this Lemma using Lemma 20.

First, we construct $\mathcal{W}$, $\mathcal{V}$, $d$, $s$, and $P_\theta$. Let $\mathcal{W} = \mathbb{S}^{d-1}$. Let $\mathcal{V}$ be the set in Lemma 22 with $\gamma = 2\epsilon$. For any $w_1, w_2 \in \mathcal{W}$, let $d(w_1, w_2) = \theta(w_1, w_2)$, $s = \pi\epsilon$. Fix any algorithm $\mathcal{A}$. For any $w \in \mathcal{W}$, any $x \in \mathcal{X}$, define $P_w[Y = 1 | X = x] = \begin{cases} 1 - \eta, & w \cdot x \ge 0 \\ \eta, & w \cdot x < 0 \end{cases}$, and $P_w[Y = 0 | X = x] = 1 - P_w[Y = 1 | X = x]$. Define $P_w^n$ to be the distribution of $n$ examples $\{(X_i, Y_i)\}_{i=1}^n$ where $Y_i$ is drawn from distribution $P_w(Y|X_i)$ and $X_i$ is drawn by the active learning algorithm $\mathcal{A}$ based solely on the knowledge of $\{(X_j, Y_j)\}_{j=1}^{i-1}$.

By Lemma 22, we have $M = |\mathcal{V}| \ge \frac{\sqrt{d}}{2} \left( \frac{1}{4\pi\epsilon} \right)^{d-1} - 1 \ge \frac{1}{4} \left( \frac{1}{4\pi\epsilon} \right)^{d-1}$, and $d(w_1, w_2) \ge 2\pi\epsilon = 2s$ for any distinct $w_1, w_2 \in \mathcal{V}$.

Clearly, for any $w \in \mathcal{W}$, if the optimal classifier is $w$, and the oracle $\mathcal{O}$ responds according to $P_w(\cdot \mid X = x)$, then it satisfies $\eta$-bounded noise condition. Therefore, to prove the lemma, it suffices to show that if $n \le \frac{d \ln \frac{1}{\epsilon}}{16(1-2\eta)^2}$, then

$$\sup_{w \in \mathcal{W}} P_w \left( d(w, \mathcal{A}(X^n, Y^n)) \ge s \right) \ge \frac{1}{2}.$$

Now, by Lemma 20,

$$\sup_{w \in \mathcal{W}} P_w^n \left( d(w, \mathcal{A}(X^n, Y^n)) \ge s \right) \ge 1 - \frac{I(V; X^n, Y^n) + \ln 2}{\ln M} \ge 1 - \frac{I(V; X^n, Y^n) + \ln 2}{(d-1) \ln \frac{1}{4\pi\epsilon} - \ln 4}.$$

It remains to show if $n = \frac{d \ln \frac{1}{\epsilon}}{16(1-2\eta)^2}$, then $I(V; X^n, Y^n) \le \frac{1}{2} \left( (d-1) \ln \frac{1}{4\pi\epsilon} - \ln 4 \right) - \ln 2$.

By the chain rule of mutual information, we have

$$I(V; X^n, Y^n) = \sum_{i=1}^n \left( I\left(V; X_i \mid X^{i-1}, Y^{i-1}\right) + I\left(V; Y_i \mid X^i, Y^{i-1}\right) \right)$$

First, we claim $V$ and $X_i$ are conditionally independent given $\{X^{i-1}, Y^{i-1}\}$, and thus $I\left(V; X_i \mid X^{i-1}, Y^{i-1}\right) = 0$. The proof for this claim is as follows. Since the selection of $X_i$ only depends on algorithm $\mathcal{A}$ and $X^{i-1}, Y^{i-1}$, for any $v_1, v_2 \in \mathcal{V}$, $\mathbb{P}\left(X_i \mid v_1, X^{i-1}, Y^{i-1}\right) =$

$\mathbb{P}\left(X_i \mid v_2, X^{i-1}, Y^{i-1}\right)$. Thus,

$$
\begin{aligned}
\mathbb{P}\left(X_i \mid X^{i-1}, Y^{i-1}\right) &= \sum_v \mathbb{P}\left(X_i, v \mid X^{i-1}, Y^{i-1}\right) \\
&= \sum_v \mathbb{P}(v)\mathbb{P}\left(X_i \mid v, X^{i-1}, Y^{i-1}\right) \\
&= \frac{1}{|\mathcal{V}|}\sum_v \mathbb{P}\left(X_i \mid v, X^{i-1}, Y^{i-1}\right) \\
&= \mathbb{P}\left(X_i \mid V, X^{i-1}, Y^{i-1}\right)
\end{aligned}
$$

Next, we show $I\left(V; Y_i \mid X^i, Y^{i-1}\right) \leq 5(1 - 2\eta)^2 \ln 2$. On one hand, since $Y_i \in \{-1, +1\}$, $I\left(V; Y_i \mid X^i, Y^{i-1}\right) \leq H\left(V \mid X^i, Y^{i-1}\right) \leq \ln 2$. where $H(\cdot|\cdot)$ is the conditional entropy.

On the other hand,

$$
\begin{aligned}
& I\left(V; Y_i \mid X^i, Y^{i-1}\right) \\
=& \mathbb{E}_{X^i, Y^i, V}\left[\ln \frac{\mathbb{P}\left(V, Y_i \mid X^i, Y^{i-1}\right)}{\mathbb{P}\left(V \mid X^i, Y^{i-1}\right)\mathbb{P}\left(Y_i \mid X^i, Y^{i-1}\right)}\right] \\
=& \mathbb{E}_{X^i, Y^i, V}\left[\ln \frac{\mathbb{P}\left(Y_i \mid V, X^i, Y^{i-1}\right)}{\mathbb{P}\left(Y_i \mid X^i, Y^{i-1}\right)}\right] \\
=& \mathbb{E}_{X^i, Y^i, V}\left[\ln \frac{\mathbb{P}\left(Y_i \mid V, X^i, Y^{i-1}\right)}{\mathbb{E}_{V'}\mathbb{P}\left(Y_i \mid V', X^i, Y^{i-1}\right)}\right] \\
\leq& \mathbb{E}_{X^i, Y^i, V, V'}\left[\ln \frac{\mathbb{P}\left(Y_i \mid V, X^i, Y^{i-1}\right)}{\mathbb{P}\left(Y_i \mid V', X^i, Y^{i-1}\right)}\right] \\
\leq& \max_{x^i, y^{i-1}, v, v'} D_{\mathrm{KL}}\left(\mathbb{P}\left(Y_i \mid x^i, y^{i-1}, v\right), \mathbb{P}\left(Y_i \mid x^i, y^{i-1}, v'\right)\right) \\
=& \max_{x^i, y^{i-1}, v, v'} D_{\mathrm{KL}}\left(\mathbb{P}\left(Y_i \mid x_i, v\right), \mathbb{P}\left(Y_i \mid x_i, v'\right)\right) \\
=& \max_{x^i, v, v'} D_{\mathrm{KL}}\left(P_v\left(Y_i \mid x_i\right), P_{v'}\left(Y_i \mid x_i'\right)\right) \\
\leq& \frac{(1 - 2\eta)^2}{\eta(1 - \eta)}
\end{aligned}
$$

where the first inequality follows from the convexity of KL-divergence, and the last inequality follows from Lemma 23.

Combining the two upper bounds, we get $I\left(V; Y_i \mid X^i, Y^{i-1}\right) \leq \min\left\{\ln 2, \frac{(1-2\eta)^2}{\eta(1-\eta)}\right\} \leq 5(1 - 2\eta)^2 \ln 2$.

Therefore, $I(V; X^n, Y^n) \leq 5n(1 - 2\eta)^2 \ln 2$. If $n \leq \frac{d \ln \frac{1}{\epsilon}}{16(1-2\eta)^2} \leq \frac{\frac{1}{2}\left((d-1)\ln \frac{1}{4\pi\epsilon} - \ln 4\right) - \ln 2}{5(1-2\eta)^2 \ln 2}$, then $I(V; X^n, Y^n) \leq \frac{1}{2}\left((d-1)\ln \frac{1}{4\pi\epsilon} - \ln 4\right) - \ln 2$. This concludes the proof. $\qquad \square$

**Lemma 25.** *For any $d > 0$, $0 \leq \eta < \frac{1}{2}$, $0 < \epsilon < \frac{1}{3}$, $0 < \delta \leq \frac{1}{4}$, for any active learning algorithm $\mathcal{A}$, there is a $u \in \mathbb{S}^{d-1}$, and a labeling oracle $\mathcal{O}$ that satisfies $\eta$-bounded noise condition with respect to $u$, such that if with probability at least $1 - \delta$, $\mathcal{A}$ makes at most $n$ queries to $\mathcal{O}$ and outputs $v \in \mathbb{S}^{d-1}$ such that $\mathbb{P}[\operatorname{sign}(v \cdot x) \neq \operatorname{sign}(u \cdot x)] \leq \epsilon$, then $n \geq \Omega\left(\frac{\eta \ln \frac{1}{\delta}}{(1-2\eta)^2}\right)$.*

*Proof.* We prove this result by reducing the hypothesis testing problem in Lemma 21 to our problem of learning halfspaces.

Fix $d, \epsilon, \delta, \eta$. Suppose $\mathcal{A}$ is an algorithm that for any $u \in \mathbb{S}^{d-1}$, under $\eta$-bounded noise condition, with probability at least $1 - \delta$ outputs $v \in \mathbb{S}^{d-1}$ such that $\mathbb{P}[\text{sign}(v \cdot x) \neq \text{sign}(u \cdot x)] \leq \epsilon < \frac{1}{3}$, which implies $\theta(v, u) \leq \frac{\pi}{3}$ under bounded noise condition.

Let $p_0 = \eta$, $p_1 = 1 - \eta$. Suppose that $\alpha \sim \text{Bernoulli}(\frac{1}{2})$ is an unknown random variable. We are given a sequence of i.i.d. (given $\alpha$) Bernoulli$(p_\alpha)$ random variables $\xi_1, \xi_2 \ldots$, and would like to test if $\alpha$ equals 0 or 1.

Define $e = (1, 0, 0, \ldots, 0) \in \mathbb{R}^d$. Construct a labeling oracle $\mathcal{O}$ such that for the $i$-th query $x_i$, it returns $2\xi_i - 1$ if $x_i \cdot e \geq 0$, and $1 - 2\xi_i$ otherwise. Clearly, the oracle $\mathcal{O}$ satisfies $\eta$-bounded noise condition with respect to underlying halfspace $u = (2\alpha - 1)e = (2\alpha - 1, 0, 0, \ldots, 0) \in \mathbb{R}^d$.

Now, we run learning algorithm $\mathcal{A}$ with oracle $\mathcal{O}$. Let $m$ be the number of queries $\mathcal{A}$ makes, and $\mathcal{A}(\xi_1, \ldots, \xi_m)$ be the normal vector of the halfspace output by the learning algorithm. We define

$$f(\xi_1, \ldots, \xi_m) = \begin{cases} 0 & \text{if } \mathcal{A}(\xi_1, \ldots, \xi_m) \cdot e < 0 \\ 1 & \text{otherwise} \end{cases}.$$

By our assumption of $\mathcal{A}$ and construction of $\mathcal{O}$, $\mathbb{P}\left(\theta\left(u, \mathcal{A}(\xi_1, \ldots, \xi_m)\right) \leq \frac{1}{3}\pi\right) \geq 1 - \delta$, so $\mathbb{P}\left(f(\xi_1, \ldots, \xi_m) = \alpha\right) \geq 1 - \delta$, implying $\mathbb{P}\left(f(\xi_1, \ldots, \xi_m) \neq \alpha\right) \leq \delta$. By Lemma 21, $m \geq 2\left\lfloor \frac{4\eta(1-\eta)}{(1-2\eta)^2} \ln \frac{1}{8\delta(1-2\delta)} \right\rfloor = \Omega\left(\frac{\eta \ln \frac{1}{\delta}}{(1-2\eta)^2}\right)$. $\qquad \square$