[Reviews · NeurIPS 2017]

Reviewer 1



Summary: This paper studies the problem of learning linear separators under the uniform distribution with noisy labels. The two noise models considered are: bounded noise where the label of each example if flipped independently with some probability upper bounded by eta < 1/2, and the adversarial noise model where some arbitrary eta fraction of the labels are flipped. For the bounded noise model previous result shows how to learn the separator to additive accuracy epsilon with time and sample complexity that is polynomial in the dimensionality d but exponential in the noise parameter (1-\eta). The authors propose a perceptron based algorithm that achieves sample complexity that is near optimal. Furthermore, the algorithm is an active learning algorithm and has label complexity logarithmic in 1/epsilon. In addition, since it is a perceptron based algorithm it can run in the streaming setting. For the adversarial noise case the authors show that the algorithm can handle noise that is epsilon/log(d) where epsilon is the desired accuracy. This is worse than the best known result but the algorithm has better run time. Techniques: The algorithm involves running the perceptron algorithm but update only if the mistake lies within a certain margin of the current guess of the separator. This margin needs to be updated after every epoch. The technical difficulty is to show that, with high probability, each epoch is getting closer to the target. This is easy to show in expectation but requires a careful martingale analysis to get a high probability bound. This is a solid theoretical paper that tackles a challenging problem of handling bounded noise and improves over the best known result. The paper is well written and the analysis techniques used should have applications to other settings as well.

Reviewer 2



This paper gives improved bounds for active learning of halfspaces (linear thresholds) under the uniform distribution on inputs and different bounded classification noise models. The improvement is either in the sample complexity or in the running time. The algorithms proposed are simple, clever variants of the perceptron method. On the positive side, it is a good step towards improving our understand of noise-tolerance and efficient active learning. The downside is that the results are specialized to the uniform distribution (this should be stated explicitly in the abstract, intro and theorem statements), since halfspaces can be learned from any distribution, and even under random and other milder noise models.

Reviewer 3



This paper analyzes a variant of the Perceptron algorithm for active learing of origin-centered halfspaces under the uniform distribution on the unit sphere. It shows that their algorithm learns using a nearly tight number of samples in the random independent noise of bounded rate. Previous work had exponentially worse dependence on the noise rate. In addition, it shows that this algorithm can deal with adversarial noise of sufficiently low rate. The latter result improves polynomially on the sample complexity but requires a stronger condtion on the noise rate. The assumptions in this setting are very strong and as a result are highly unlikely to hold in any realistic problem. At the same time there has been a significant amount of prior theoretical work on this and related settings since it is essentially the only setting where active learning gives an exponential improvement over passive learning in high dimensions. This work improves on some aspects of several of these works (although it is somewhat more restrictive in some other aspects). Another nice feature of this work is that it relies on a simple variant of the classical Perceptron algorithm and hence can be seen as analyzing a practically relevant algorithm in a simplified scenario. So overall this is solid theoretical contribution to a line of work that aims to better understand the power of active learning (albeit in a very restricted setting). A weakness of this work is that beyond the technical analysis it does not have much of an algorithmic or conceptual message. So while interesting to several experts it's unlikely to be of broader interest at NIPS. ---response to rebuttal --- >> "This label query strategy is different from uncertainty sampling methods in practice, which always query for the point that is closest to the boundary. We believe that our novel sampling strategy sheds lights on the design of other practical active learning algorithms as well." This is an interesting claim, but I do not see any support for this in the paper. The strategy might be just an artifact of the analysis even in your restricted setting (let alone a more practical one). So my evaluation remains positive about the technical achievement but skeptical about this work making a step toward more realistic scenarios.